# DIFFERENTIALLY PRIVATE VISION-LANGUAGE FOUNDATION MODELS VIA IMAGE CAPTIONING

## ABSTRACT

The common practice of training foundation models on web-crawled data raises privacy and copyright concerns, as sensitive training data can be memorized by the model and unintentionally misused. Differential privacy (DP) is a robust and rigorous framework for mitigating against such risks, albeit often with significant performance loss and is commonly perceived as unviable in most use cases. In this work, we demonstrate that combining DP with vision-language pre-training can be a powerful recipe for obtaining differentially private foundation models trained *from scratch*. Our model uses text supervision to learn superior image representations, and also exhibits the first instance of multi-modal capabilities for DP training. Under a privacy budget of $\varepsilon = 8$, our image captioner (DP-Cap) trained on a 233M subset of the LAION-2B dataset attains *54.4% zero-shot accuracy* on CIFAR-10. On the challenging ARO benchmark, DP-Cap achieves performance close to its non-private counterpart (Cap), and greatly surpasses the best *non-private* CLIP model. Our work challenges the prevailing sentiment that high-utility foundation models are unattainable for DP training from scratch.

## 1 INTRODUCTION

The field of artificial intelligence in recent years has shifted dramatically towards the paradigm of pre-training so-called *foundation models* (Brown et al., 2020a; Oquab et al., 2023; Touvron et al., 2023) on massive internet-scale datasets. These models are capable of learning complex relationships between concepts and entities from multiple modalities such as image, text, video and speech (Girdhar et al., 2023), and serve as the backbone for applications to downstream tasks via transfer learning. Unfortunately, this paradigm shift has introduced new security and privacy risks as well (Bommasani et al., 2021). In particular, training data for foundation models may consist of data that contain copyrighted material or personal identifiable information about individuals (Henderson et al., 2023). Combined with the fact that these models are prone to instance-level memorization (Zhang et al., 2021) and can even regurgitate their training data unintentionally (Carlini et al., 2019; 2021b), their use can implicate copyright and privacy concerns when doing so without care. Indeed, several ongoing lawsuits have already challenged the legality of training foundation models on internet data (Henderson et al., 2023).

Given these concerns, it is important to design training procedures for foundation models that can reduce unintended memorization. Among current candidate solutions, differential privacy (DP; Dwork et al. (2006)) stands out as a robust and mathematically rigorous framework for protecting training data privacy. Henderson et al. (2023) has also suggested DP as a legally viable mitigation against training data copyright violation. The quintessential DP training algorithm — DP-SGD (Song et al., 2013; Abadi et al., 2016) — has been applied to train large-scale image classification models (De et al., 2022) and language models (Anil et al., 2021; Mehta et al., 2022). More recently, Yu et al. (2023) successfully trained the first vision foundation model on the LAION-400M dataset (Schuhmann et al., 2021) in a self-supervised manner, with the motivation that scaling DP training to large-scale internet data is important in achieving better utility with reasonable privacy guarantees.

However, in almost all application settings, DP training results in a model that suffers a large utility drop when aiming to attain a strong privacy guarantee (*e.g.*, $\varepsilon \leq 10$), rendering it generally unviable. The core difficulty in DP training of foundation models is representation learning. For example, Tramer & Boneh (2020) showed that DP-trained models often produce representations that are worse

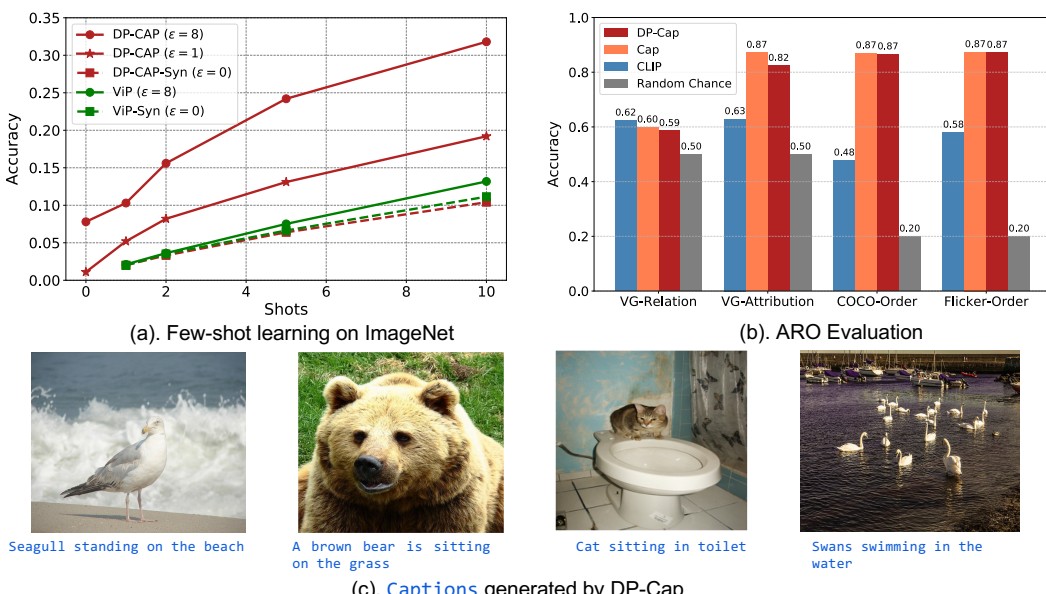

(a). Few-shot learning on ImageNet

(b). ARO Evaluation

(c). Captions generated by DP-Cap

Figure 1: (a) Few-shot ImageNet-1K linear prob accuracy comparison between DP-Cap (ours) and ViP (Yu et al., 2023). DP-Cap learns better image representations using the same training data and privacy budget, and considerably surpasses synthetic initialization (*syn*). (b) Compositional understanding evaluation on the ARO benchmark (Yuksekgonul et al., 2022). DP-Cap performance is close to non-private Cap and outperforms non-private CLIP. (c) Captions generated by DP-Cap on images from the MS-COCO 2017 (Lin et al., 2015) test set.

than handcrafted ones, and when given a good pre-trained model, DP fine-tuning can in fact reach close to non-private model performance (De et al., 2022; He et al., 2022a).

To address this challenge, we demonstrate that vision-language pre-training is a promising strategy to enhance the representation learning abilities of DP foundation models. Using both modalities, this approach can better leverage the information from each sample by extracting a denser signal when compared to training on images only. Specifically, we opt for image captioning as we find that it is especially well aligned with the prerequisites of DP-SGD. Applying this method on a 233M subset of the LAION-2B dataset, we train *the first DP vision-language foundation model*, with significantly better image representations compared to image-only DP pre-training on the same dataset and under the same privacy budget. As depicted in Figure 1(a), our model trained with a privacy budget of $\varepsilon = 8$, referred to as "DP-Cap", shows substantial improvements in few-shot ImageNet-1K linear prob accuracy compared to the ViP model (Yu et al., 2023), both trained on the same dataset. We also demonstrate that DP-Cap can achieve competitive performance even under $\varepsilon = 1$.

DP-Cap also exhibits strong multimodal capabilities, the first occurrence for DP-trained models. In Figure 1(b), we evaluate the compositional understanding capabilities of DP-Cap on the challenging ARO (Attribution, Relation, and Order) benchmark (Yuksekgonul et al., 2022) and show that it attains significantly better performance compared to the *non-private* CLIP model (Radford et al., 2021), presented as a reference point. As a qualitative evaluation, we use the trained DP-Cap model to caption several images from the MS-COCO 2017 (Lin et al., 2015) test set in Figure 1(c) and Appendix B.3. The resulting captions are grammatically correct and semantically coherent, while (close to) accurately describing contents of the image; this is interesting because our model has only been exposed to language supervision from LAION, which are far from being flawless. Our results suggest that training differentially private vision-language models on internet-scale datasets can be a viable approach for obtaining high-utility DP foundation models with multimodal capabilities.

## 2 BACKGROUND AND RELATED WORK

### 2.1 SELF-SUPERVISED AND VISION-LANGUAGE PRE-TRAINING

**Self-supervised learning (SSL)** is a learning paradigm where a model is trained using only unlabeled data. In doing so, the model learns generic and transferable data representations that can be used in downstream tasks via fine-tuning. Most of the foundation models, such as BERT (Devlin et al., 2018) and DINO (Caron et al., 2021; Oquab et al., 2023), are trained with self-supervised

learning methods (Bommasani et al., 2021). For image representation learning, the most popular category of SSL methods is contrastive learning (Chen & He, 2021; He et al., 2020; Chen et al., 2020b). Here, the learning objective ensures that different augmented views (*i.e.*, with the same semantic information) of each image have similar representations, while enforcing dissimilarity between the representations of distinct images. Other families of SSL methods include clustering-based methods (Caron et al., 2020; 2021), covariance regularization (Zbontar et al., 2021; Ermolov et al., 2021; Bardes et al., 2021) and reconstruction (He et al., 2022b; Geng et al., 2022).

**Vision-language pre-training.** Many modern ML datasets such as Conceptual Captions (Changpinyo et al., 2021), LAION (Schuhmann et al., 2021) and DataComp (Gadre et al., 2023) consist of aligned image-text pairs where the image and text contain roughly similar semantic information. One can leverage the aligned nature of the training data to pre-train *vision-language models* (VLMs) that connect the two modalities and perform more general multimodal tasks. Contrastive learning-based techniques such as CLIP (Radford et al., 2021) and BLIP (Li et al., 2022) are also applicable for pre-training vision-language foundation models. Doing so not only learns high-quality image and text representations but also introduces new multimodal capabilities such as cross-modal retrieval and zero-shot prediction (Radford et al., 2021). Recent work by Tschannen et al. (2023) shows an image captioning approach (predicting text captions from images) is a viable alternative to contrastive learning and can lead to models with robust performance.

## 2.2 DIFFERENTIAL PRIVACY AND DP-SGD

In the following, we denote by $\mathcal{M}$ a randomized learning algorithm, which takes a dataset $\mathcal{D}$ containing $N$ samples and produces a machine learning model $\boldsymbol{\theta}$ through the process $\mathcal{M}(\mathcal{D})$.

**Differential privacy** (Dwork et al., 2006). A randomized mechanism $\mathcal{M}$ is $(\varepsilon, \delta)$-DP if, for any two adjacent datasets $\mathcal{D}$ and $\mathcal{D}'$ differing by a single sample, and for any subset $\mathcal{O} \subset \mathbf{Im}(\mathcal{M})$:

$$\mathbf{P}[\mathcal{M}(\mathcal{D}) \in \mathcal{O}] \leq \mathbf{P}[\mathcal{M}(\mathcal{D}') \in \mathcal{O}] \exp(\varepsilon) + \delta. \tag{1}$$

We adopt the leave-one-out notion of adjacency in this work, *i.e.*, $\mathcal{D} = \mathcal{D}' \cup \{\mathbf{x}\}$ for some sample $\mathbf{x}$ or vice versa. The protection offered by differential privacy bounds the extent to which any potential adversary can infer information about the dataset $\mathcal{D}$ after observing the algorithm's output. In the context of ML, this implies that if we obtain the model $\boldsymbol{\theta}$ through a DP training algorithm $\mathcal{M}$ then its training data is provably difficult to recover or infer (Balle et al., 2022; Guo et al., 2022; 2023).

**DP-SGD** (Song et al., 2013; Abadi et al., 2016) has emerged as the predominant differentially private algorithm for training deep neural networks (DNNs). At each gradient step $k$, a batch $\mathcal{B}_k$ is sampled where each example from the training data is chosen randomly with probability $q = B/N$, where $B$ represents the average batch size. For $C > 0$, define the clipping function for any $X \in \mathbb{R}^d$ by $\text{clip}_C(X) = C \cdot X/\|X\|$ if $\|X\| \geq C$ and $\text{clip}_C(X) = X$ otherwise. Given model parameters $\boldsymbol{\theta}_k$, DP-SGD defines the update $\boldsymbol{\theta}_{k+1} = \boldsymbol{\theta}_k - \eta_k \widetilde{\mathbf{g}}_k$ where $\eta_k$ is the step size and $\widetilde{\mathbf{g}}_k$ is given by:

$$\widetilde{\mathbf{g}}_k := \frac{1}{B} \left[ \sum_{i \in \mathcal{B}_k} \text{clip}_C \left( \nabla_{\boldsymbol{\theta}} \ell_i(\boldsymbol{\theta}_k) \right) + \mathcal{N} \left( 0, C^2 \sigma^2 \mathbf{I} \right) \right], \tag{2}$$

where $\ell_i(\boldsymbol{\theta})$ is the per-sample loss function evaluated at sample $\mathbf{x}_i$. We also use the term "DP-SGD" loosely to refer to the category of gradient-based optimization algorithms that operate on the noisy gradient, *e.g.*, Adam (Kingma & Ba, 2014). The privacy analysis of DP-SGD relies on composition of multiple steps. One particularly powerful analysis framework amenable to such compositions relies on a variant of DP called Rényi differential privacy (RDP) (Mironov, 2017). An advantage of RDP is its additive composition property, where the privacy guarantees of a sequence of mechanisms can be combined with amplification by subsampling (Wang et al., 2019) and then translated to $(\varepsilon, \delta)$-DP (Balle et al., 2020; Gopi et al., 2021). In this work, we adopt this accounting technique.

**Scaling up DP-SGD training.** DP training is a theoretically and empirically proven remedy against unintended training data memorization. Even models with large $\varepsilon$ (*e.g.*, $\varepsilon = 100$) can empirically defend against privacy attacks (Carlini et al., 2021a; Guo et al., 2023). Despite its great appeal, DP training also carries a significant drawback of large drop in model utility (Abadi et al., 2016; Tramer & Boneh, 2020). For example, the SOTA performance on ImageNet when training from scratch with a DP guarantee of $\varepsilon = 8$ is 39.2% (Sander et al., 2023); in comparison, the non-private performance on ImageNet when training from scratch can reach 88% (Touvron et al., 2022) or higher.

So far the best remedy for this substantial performance drop is relying on a pre-trained foundation model. It then only needs to be fine-tuned with DP-SGD on the private dataset, requiring to extract much less information from the training samples in order to achieve close to the non-private performance (Li et al., 2021a; Berrada et al., 2023; De et al., 2022). However, it shifts the responsibility of protecting data privacy to the foundation model training. The need for DP pre-training thus arises from the fact that what is considered "public" data may not always truly be public.

More recently, without utilizing any real-world samples as "*public pre-training dataset*", Yu et al. (2023) made the first step towards training foundation models with differential privacy. The authors observed that contrastive learning cannot be applied in a nutshell using DP-SGD (*cf.* equation 2), and instead opted for the reconstruction-based approach of masked autoencoder—MAE (He et al., 2022b). By leveraging weight initialization through synthetic pre-training, the authors were able to obtain high-utility vision foundation models at a strict privacy budget of $\varepsilon = 8$. Compared to ViP (Yu et al., 2023), we demonstrate that the image captioning approach (see Section 3.1) learns much better visual representations by utilizing the additional text supervision.

## 3 PRE-TRAINING DP IMAGE CAPTIONING MODELS

We describe in detail our approach of DP vision-language pre-training via image captioning. We first argue why image captioning is intuitively a suitable objective for obtaining better image representations via DP-SGD training (section 3.1). Then, using ablation studies, we demonstrate the specific technical challenges that we resolved to make DP training viable for image captioning (section 3.2).

### 3.1 PRE-TRAINING IMAGE CAPTIONERS WITH DP-SGD

**Why is DP representation learning difficult?** Representation learning lies at the core of deep learning, empowering models to extract meaningful features from raw data. However, in the context of DP training, representation learning becomes very difficult. Intuitively, DP restricts the amount of information each sample can contribute to the model via the privacy parameter $\varepsilon$. Thus, to learn good representations, DP training requires much more data compared to non-private training. This intuition is corroborated by empirical findings that DP training on small datasets learn worse representations than handcrafted ones (Tramer & Boneh, 2020), and given good representations, DP fine-tuning can reach close to non-private model performance (De et al., 2022; He et al., 2022a).

To obtain high-quality learned representations via DP training, it is thus necessary to tap into massive internet-scale uncurated datasets. Indeed, Yu et al. (2023) showed that by pre-training with DP-SGD a masked autoencoder (MAE; He et al. (2022b)) objective, it is possible to obtain image representations that rival AlexNet (Krizhevsky et al., 2012) in quality. However, the MAE objective encourages the model to learn extraneous image details for reconstruction and does not scale well in terms of batch size — a crucial factor for reducing the effective noise in DP training — limiting its potential for scaling up DP training; we will demonstrate this fact in Section 3.2.

**Why is DP vision-language pre-training suitable?** Given image-text aligned datasets, pre-training using language supervision becomes an appealing option for both non-private and private learning. Compared to image-only supervision, language supervision contains a more condensed summary of the image content, allowing the model to ignore irrelevant details such as background and focus on objects of interest and their relationships. This is especially helpful for DP since the model needs to extract as much useful information as possible from each sample given the privacy budget $\varepsilon$.

In addition, we show that vision-language pre-training supports a very large batch size, much larger than what is typically used in image-only pre-training (Radford et al., 2021; Li et al., 2022; Yu et al., 2022). This subtle aspect is in fact crucial for reducing the effective noise in DP-SGD, which allows the model parameters to converge to a stable solution with lower training loss (see Section 3.2).

**Vision-language pre-training via image captioning.** Perhaps the most popular approach for training vision-language models is contrastive language image pre-training (CLIP) (Radford et al., 2021) as well as its variants (Mu et al., 2022; Li et al., 2023). However, the contrastive loss used in these methods is not an additive function over the samples, *i.e.*, it cannot be written in the form $\sum_i \ell_i$, where $\ell_i$ depends only on the $i$-th sample. Thus, DP-SGD (*cf.* equation 2) cannot directly be used.

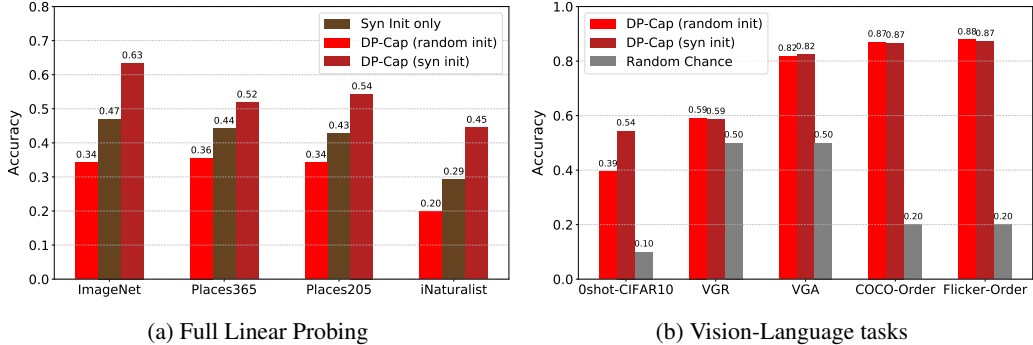

(a) Full Linear Probing  (b) Vision-Language tasks

Figure 2: **Impact of synthetic initialization on the DP-Cap model.** The learned image representation benefits substantially from initializing on the Shaders21k dataset. The accuracy gap between DP-Cap (random init) and DP-Cap (syn init) can be as large as 24% for ImageNet linear probing and 21% for CIFAR-10 zero-shot prediction.

Unlike contrastive learning, the image captioning approach (Sariyildiz et al., 2020; Desai & Johnson, 2021; Tschannen et al., 2023) aligns particularly well with DP-SGD training. Specifically, an image captioner is a model trained to predict captions based on their corresponding images. The training objective of the image captioner for one image-text pair $[\mathbf{x}^{\text{img}}, \mathbf{z}^{\text{text}}]$ is to minimize over $\boldsymbol{\theta} := \{\boldsymbol{\theta}_{\text{enc}}, \boldsymbol{\theta}_{\text{dec}}\}$ the following loss:

$$L_{\text{Cap}}([\mathbf{x}^{\text{img}}, \mathbf{z}^{\text{text}}]; \boldsymbol{\theta}) := \frac{1}{T} \sum_{t=0}^{T-1} \ell_{\text{CE}}\Big( z_{t+1}^{\text{text}}, \varphi\big( \underbrace{\psi(\mathbf{x}^{\text{img}}; \boldsymbol{\theta}_{\text{enc}})}_{\text{image embedding}}, \underbrace{z_1^{\text{text}}, \dots, z_t^{\text{text}}}_{\text{first } t \text{ tokens}}; \boldsymbol{\theta}_{\text{dec}}\big)\Big), \qquad (3)$$

where $\mathbf{z}^{\text{text}}$ denotes the caption token sequence $\{z_1^{\text{text}}, \dots, z_T^{\text{text}}\}$ and the image captioner consists of two parts: the image encoder $\psi(\cdot; \boldsymbol{\theta}_{\text{enc}})$ and the text decoder $\varphi(\cdot; \boldsymbol{\theta}_{\text{dec}})$. The rationale behind the design of equation 3 is that the image encoder maps the input image $\mathbf{x}^{\text{img}}$ to an embedding vector, and the text decoder takes the image embedding $\psi(\mathbf{x}^{\text{img}}; \boldsymbol{\theta}_{\text{enc}})$ and the first $t$ caption tokens $\{z_1^{\text{text}}, \dots, z_t^{\text{text}}\}$ as inputs and predicts the next caption token $z_{t+1}^{\text{text}}$. Both Encoder and Decoder are optimized to maximize the log-likelihood of the correct next token. Equation 3 corresponds to the loss function for an image-text pair $[\mathbf{x}^{\text{img}}, \mathbf{z}^{\text{text}}]$; summing over all the samples in a batch gives the complete empirical loss in an additive form, which can directly be used with DP-SGD.

### 3.2 Unleashing the Efficacy of Differentially Private Image Captioning

Although image captioning has demonstrated impressive representation learning capabilities in the non-private regime, adapting it to DP training requires careful considerations. To obtain a useful pre-trained model, one needs to train for a sufficient number of steps under a low effective noise, both of which are at odds with obtaining a strong privacy guarantee. In the following, we detail the strategy we used to handle this trade-off when training the image captioner.

**Sufficient number of training steps.** We address the challenge of training for a sufficient number of steps via synthetic pre-training. As shown in Yu et al. (2023), synthetic images consisting of only textures can provide a good initialization for the model without any privacy risk. With this better initialization, the model can focus on learning dataset-specific properties rather than low-level image properties such as edge detectors, therefore expending privacy budget in a more optimal manner. We adapt this technique (by pre-training on the Shaders21k dataset (Baradad et al., 2022) dataset; see Appendix A.1) for initializion, and observe that it is even more effective with DP-Cap compared to ViP. As shown in Fig. 2, our *DP-Cap (syn init)* improves over *DP-Cap (random init)* by more than 24% on ImageNet-1k linear probing. It also improves over synthetic initialization alone (*Syn-init*) by more than 14% on ImageNet-1k linear probing, whereas the gain in ViP is smaller than 6%.

**Low effective noise $\sigma/B$.** In addition to the better initialization, the model also needs to be trained under a small amount of DP noise to facilitate convergence. It is easy to verify that the effective noise added to the average gradient has magnitude $\sigma/B$ (*cf.* equation 2). Thus, one needs to either reduce $\sigma$ or increase $B$ to lower the effective noise. While decreasing $\sigma$ may seem appealing, RDP account techniques predict that when $\sigma$ decreases to values close to 0.5, $\varepsilon$ undergoes a substantial increase (Bun & Steinke, 2016; Dwork & Rothblum, 2016; Mironov et al., 2019; Sander et al., 2023)

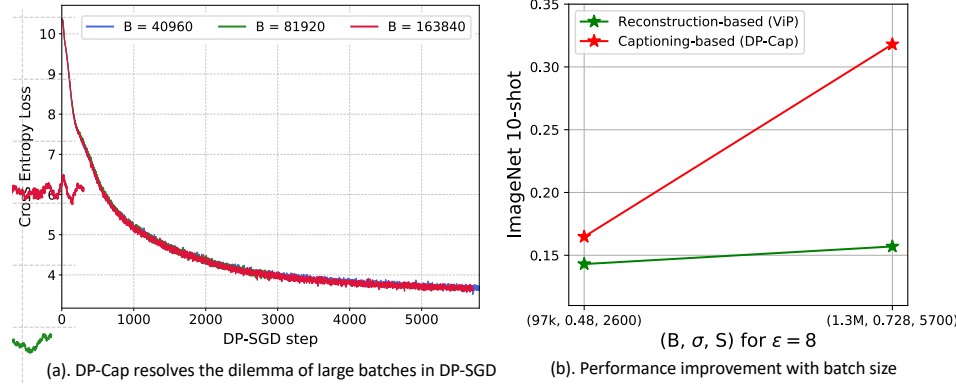

Figure 3: (a) We fix the effective noise per step $\sigma/B = 5.6 \times 10^{-7}$ (corresponding to our set of parameters) and show that the loss trajectory is remarkably consistent across different batch sizes, allowing us to effectively scale up batch size to reduce the effective noise. (b) Performance from two sets of parameters that provide $\varepsilon = 8$: with batch size 90k (used to train ViP (Yu et al., 2023)), and the 1.3M batch size. Contrary to ViP, DP-Cap successfully leverages the better SNR.

For instance, on LAION-233M, training for 26,756 steps with a batch size of 1.3M and $\sigma = 1.0$ results in $\varepsilon = 8$. To halve the effective noise ($\sigma/B$) per step, we can double the batch size $B$, which would lead to running DP-SGD for 6,500 steps to obtain the same $\varepsilon$. In comparison, if we instead halve $\sigma$, *i.e.*, $\sigma = 0.5$, and hold everything else constant, RDP accounting for $\varepsilon = 8$ would only permit *eight* gradient steps! In this scenario, the benefit of having more data diminishes because a significant reduction in the number of training steps is required to maintain a fixed privacy budget. As a result, increasing the batch size $B$ is a much more effective strategy than reducing $\sigma$ (beyond a certain threshold) to decrease the effective noise. See Section B.2 for further discussion.

**Resolving the dilemma of large batch training in DP-SGD.** Larger batch leads to a smaller effective noise in DP-SGD. At the same time, it is difficult to employ extremely large batch sizes (*e.g.* $B = 1M$) for both private and non-private model training in practice. Specifically, Sander et al. (2023) observed that when training an image classifier from scratch with DP-SGD on ImageNet, at a fixed number of steps $S$ and fixed effective noise $\sigma/B$, the performance decreases significantly w.r.t. the batch size: in particular for $B \in [128, 16384]$, a drop of 10% in top-1 accuracy was observed. We refer to this as the *dilemma of large batch training in DP-SGD*, which drastically limits the power of the previous observation, *i.e.* that increasing B reduces the effective noise.

Intriguingly, we find that vision-language pre-training can partially resolve this dilemma by allowing larger batch training. In Figure 3(a), we compare the loss behaviors when scaling the batch size for DP-Cap. We fix the effective noise $\sigma/B$ while varying the batch size. In stark contrast to the previous observation from Sander et al. (2023), the loss trajectory is identical across different batch sizes. With this observation, we are able to successfully scale up the batch size for DP-Cap to as large as $B = 1.3M$, achieving an effective noise of $5.6 \times 10^{-7}$, almost 10 times smaller than the effective noise of ViP in Yu et al. (2023). For ViP, we show in Figure 3(b) that training under the same small effective noise does not significantly improve the model performance (see Appendix A for additional numbers). The success of DP-Cap is thus primarily attributed to its remarkable ability to leverage the small effective noised induced by extremely large batch sizes.

**Efficient hyperparameter search via TAN.** Using an extremely large batch size and pre-training for a sufficient number of steps creates a difficulty for rapid experimentation and hyperparameter search, as the amount of compute required for gradient computation becomes the main bottleneck. Fortunately, the Total Amount of Noise (TAN) simulation concept introduced by Sander et al. (2023) offers an ideal solution: Instead of executing $S$ steps with parameters $(B, \sigma)$, which can require substantial computational resources but ensures good privacy guarantees, one can simulate an "equivalent" training by undertaking $S$ steps with parameters $(B/m, \sigma/m)$, where $m$ is a chosen constant. This reduces the computational overhead (*i.e.*, the number of per-sample gradient computation) by $m$. Notably, the scaling behavior of DP-Cap with respect to the batch size makes it highly compatible with this simulation technique. In our experiments, we employed a compute budget of 32 epochs for the final training runs of DP-Cap, and used TAN with $m = 32$ to reduce the compute to one epoch for optimizer and other hyperparameters search. This allowed us to reliably predict the performance of DP-Cap under different design choices using much less compute.

Table 1: Linear probing evaluation on downstream classification. Results for DP-NFNet, TAN, AlexNet and SimCLR are obtained from Yu et al. (2023). For ViP, we train with the same parameters as in Yu et al. (2023) but on the deduplicated dataset. More details are given in Appendix A.

| Model | pretraining data | DP? | ImageNet-1K | Places-365 | Places-205 | iNat-2021 |
|---|---|---|---|---|---|---|
| DP-NFNet | ImageNet-1K | ✓ | 45.3% | 40.1% | 39.2% | 28.2% |
| TAN | ImageNet-1K | ✓ | 49.0% | 40.5% | 38.2% | 31.7% |
| AlexNet | ImageNet-1K | ✗ | 56.5% | 39.8% | 35.1% | 23.7% |
| SimCLR | ImageNet-1K | ✗ | 67.5% | 46.8% | 49.3% | 34.8% |
| Cap | Dedup-LAION-233M | ✗ | 77.5% | 56.3% | 63.9% | 63.9% |
| ViP | Dedup-LAION-233M | ✓ | 55.8% | 47.7% | 49.2% | 38.2% |
| DP-Cap | Dedup-LAION-233M | ✓ | 63.4% | 51.9% | 54.3% | 44.5% |

## 4 EVALUATIONS

In this section, we present comprehensive evaluations for our DP-Cap models. We consider both vision (V) and vision-language (V-L) downstream tasks, including linear probing (V) using the vision encoder, zero-shot image classification (V-L), ARO (Attribution, Relation, and Order) (V-L), and image captioning (V-L) using vision encoder and text decoder.

### 4.1 EXPERIMENTAL SETUP

We first present the experimental setup; refer to Appendix A for additional details.

**Datasets.** Following the approach introduced by Yu et al. (2023), we first pre-train on the Shader21k dataset (Baradad et al., 2022) of synthetic images. We then train with DP on a subset comprising 233 million deduplicated (using SemDeDup (Abbas et al., 2023)), NSFW-filtered image-description pairs from the (English-only) LAION-2B dataset (Schuhmann et al., 2022), with faces that are blurred in a similar manner as Yang et al. (2021). We refer to this dataset as Dedup-LAION-233M in the remainder of the paper.

We use ImageNet-1K (Deng et al., 2009; Russakovsky et al., 2014), CIFAR-10/100 (Krizhevsky et al., 2009), Places-365/205 (Zhou et al., 2014) and iNaturalist-2021 (Van Horn et al., 2021) classification datasets to assess the performance of our vision backbone via few-shot linear probing, full linear probing and zero-shot prediction. For vision-language tasks, we employ the Visual Genome Attribution (VGA), Visual Genome Relation (VGR), COCO-order (Lin et al., 2015) and Flickr-30k (Plummer et al., 2016) datasets from the ARO benchmark (Yuksekgonul et al., 2022). These datasets evaluate the model's ability to understand the compositional relationship between objects and attributes, such as distinguishing between "the horse is eating the grass" and "the grass is eating the horse". We also evaluate image captioning using the MS-COCO 2017 (Lin et al., 2015) test set.

**Model and training.** Following the setup in Tschannen et al. (2023), we use a transformer architecture (Vaswani et al., 2017) for both the encoder and the decoder of DP-Cap, where the decoder applies causal cross-attention; see Section A.1 and Tschannen et al. (2023) for details. For privacy accounting we use the technique from Balle et al. (2020) along with privacy amplification via Poisson subsampling (Wang et al., 2019) through the PyTorch-based Opacus library (Yousefpour et al., 2021), targeting $\delta = 1/N$ where $N$ represents the number of training samples. We refer to the non-private counterpart of DP-Cap trained on the same Dedup-LAION-233M dataset as "Cap".

### 4.2 EVALUATION ON VISION (V) AND VISION-LANGUAGE (V-L) TASKS

**Linear probing evaluation (V).** We assess the performance of the vision encoder on downstream tasks via linear probing. In Fig 1(a), we compare the performance of DP-Cap and ViP (Yu et al., 2023) on ImageNet-1k few-shot linear probing. DP-Cap significantly improves over ViP, with up to ×2.5 better performance across different shots. In addition, we evaluate the full linear probing accuracy of DP-Cap, ViP and other baselines in Table 1. DP-Cap outperforms ViP and other DP models, including TAN (Sander et al., 2023) and DP-NFNet (De et al., 2022), across all tasks. DP-Cap even outperforms non-private AlexNet (Krizhevsky et al., 2012) and except on ImageNet, SimCLR (Chen et al., 2020a) (both were trained on ImageNet).

Table 2: Zero-shot prediction and compositional understanding (ARO). CLIP's zero-shot results are obtained from Radford et al. (2021) (base model). For ARO, see Appendix A.2.2.

| Model | DP? | Zero-shot | | | ARO | | | |
| --- | --- | --- | --- | --- | --- | --- | --- | --- |
| | | ImageNet-1k | CIFAR10 | CIFAR100 | VGR | VGA | COCO | Flickr |
| Random Chance | - | 0.1% | 10% | 1% | 50% | 50% | 20% | 20% |
| CLIP | ✗ | 62.2% | 91.3% | 65.1% | 62.4% | 62.9% | 47.8% | 58.0% |
| Cap | ✗ | 25.2% | 90.0% | 37.4% | 59.9% | 87.2% | 87.0% | 87.4% |
| DP-Cap | ✓ | 7.8% | 54.4% | 16.4% | 58.6% | 82.4% | 86.6% | 87.2% |

**Zero-shot performance (V-L)** refers to the model's ability to perform a task it has never been explicitly trained on, and is one of the most widely used metrics for evaluating vision-language models (Radford et al., 2021). Given an image, we perform zero-shot classification using DP-Cap by evaluating the likelihood of captions of the form "this is a photo of a [label]". We enumerate over different labels and predict the class that has the highest likelihood; see Section A.2.1 for full details.

In the left three columns of Table 2, we evaluate the zero-shot performance of DP-Cap compared to non-private Cap and CLIP/BLIP on ImageNet-1k and CIFAR10/100. Contrastive methods such as CLIP and BLIP have demonstrated greater suitability for zero-shot prediction compared to image captioning approaches (Tschannen et al., 2023), which is evident by the disparity between the performance of Cap and CLIP/BLIP. Nevertheless, we observe that DP-Cap achieves noteworthy zero-shot classification performance that is significantly above random chance, and stands as the first DP model to do so. This accomplishment marks a promising milestone for DP training, although there remains a substantial performance gap between DP-Cap and Cap.

**Attribution, Relation, and Order (ARO) evaluation (V-L).** The ARO benchmark, introduced by Yuksekgonul et al. (2022), serves as an evaluation framework to gauge the adeptness of VLMs in understanding the compositional relationship between objects and attributes. As observed in prior work (Yuksekgonul et al., 2022; Tejankar et al., 2021; Basu et al., 2023), contrastive-based methods such as CLIP often exhibit behavior akin to bag-of-words models, making them less adept at performing well on these benchmarks. In contrast, as shown in previous work (Tschannen et al., 2023), image captioning based models such as Cap showcase superior performance compared to CLIP. Remarkably, DP-Cap significantly outperforms CLIP in this context (see Fig 1(b) and Table 2), and even achieves performance close to that of non-private Cap. Our result shows that DP training can be particularly effective for learning complex compositional relationships.

### 4.3 SCALING BEHAVIOR OF DP-CAP

**Scaling dataset size.** We first show that dataset scaling is crucial for effectively training DP-Cap as it results in better SNR under the same privacy budget (see Figure 4 in Appendix B.2). We randomly subsample 1% and 10% of the Dedup-LAION-233M dataset, which is used for training our default DP-Cap-Base model in Table 1, and we denote these two datasets by Dedup-LAION-2M and Dedup-LAION-23M. We set the batch size to $B/100$ for Dedup-LAION-2M and $B/10$ for Dedup-LAION-23M, respectively. This allows the model to be trained for the same number of steps across the different datasets, although at a much larger effective noise level. As shown in Table 3, the number of training samples is critical for achieving strong performance for DP-Cap models: the zero-shot performance of our model trained on 1% of the dataset achieves random zero-shot performance on ImageNet and much worse accuracy across the board on ARO.

**Impact of the privacy budget $\varepsilon$.** We also investigate the performance of DP-Cap under lower privacy budgets ($\varepsilon = 1$ and $\varepsilon = 2$), employing the same batch size of 1.3 million. The outcomes of these experiments are presented in Table 3. As anticipated, the utility of our models does exhibit a decline with decreasing $\varepsilon$. However, the performance degradation is relatively minor for the vision backbone, with 10-shot ImageNet performance decreasing from 25.7% ($\varepsilon = 8$) to 19.9% ($\varepsilon = 1$). More surprisingly, the performance impact on ARO is nearly negligible. It is noteworthy that both models continue to outperform previous state-of-the-art DP models trained with $\varepsilon = 8$ (see Figure 1). This phenomenon can be attributed to the relatively small effective noise resulting from the extreme batch size, which for $\varepsilon = 1$ remains five times smaller than that used in Yu et al. (2023).

Table 3: Ablation studies on the effect of dataset size and privacy budget $\varepsilon$ on DP-Cap (base).

| $\varepsilon$ | $\sigma$ | # Data | # Steps | $B$ | ImageNet-1K | | | | ARO (V-L) | | | |
|---|---|---|---|---|---|---|---|---|---|---|---|---|
| | | | | | 0-shot (V-L) | 1-shot (V) | 2-shot (V) | 10-shot (V) | VGR | VGA | COCO | Flickr |
| $+\infty$ | 0 | 233M | 60,000 | 40,960 | 25.2% | 27.0% | 37.2% | 57.9% | 59.9% | 87.2% | 87.0% | 87.4% |
| 8.0 | 0.728 | 233M | 5708 | 1.3M | 7.8% | 10.3% | 15.6% | 31.8% | 58.6% | 82.4% | 86.6% | 87.2% |
| 2.0 | 1.18 | 233M | 2854 | 1.3M | 3.2% | 7.0% | 10.8% | 23.9% | 58.5% | 79.7% | 85.3% | 86.6% |
| 1.0 | 1.5 | 233M | 1427 | 1.3M | 1.1% | 5.2% | 8.2% | 13.1% | 58.3% | 75.6% | 83.9% | 85.3% |
| 8.0 | 0.728 | 23M | 5708 | 130K | 0.7% | 3.4% | 5.3% | 13.3% | 58.3% | 76.2% | 84.9% | 85.9% |
| 8.0 | 0.728 | 2.3M | 5708 | 13K | 0.1% | 1.8% | 2.9% | 8.1% | 57.6% | 66.4% | 79.5% | 82.0% |

Table 4: Ablation studies on the effect of model size. We compare ViP and DP-Cap's number of encoder parameters. More details about the DP-Cap models can be found in Table 5.

| Model | Config | # parameters | ImageNet-1K (Vision) | | | | ARO (Vision-Language) | | | |
|---|---|---|---|---|---|---|---|---|---|---|
| | | | 1-shot | 2-shot | 5-shot | 10-shot | VGR | VGA | COCO | Flickr |
| *ViP* | Base | 86.6M | 2.5% | 4.2% | 8.5% | 14.3% | / | / | / | / |
| *DP-Cap* | Tiny | 22.0M | 7.9% | 12.1% | 18.7% | 25.2% | 58.6% | 79.1% | 85.7% | 87.1% |
| *DP-Cap* | Small | 49.0M | 9.0% | 14.0% | 21.6% | 28.9% | 59.1% | 80.5% | 86.0% | 86.6% |
| *DP-Cap* | Base | 86.6M | 10.3% | 15.6% | 24.2% | 31.8% | 58.6% | 82.4% | 86.6% | 87.2% |

**Scaling model size.** Scaling up the model size is one of the most effective approaches for training better non-private foundation models (Brown et al., 2020b; Bommasani et al., 2021; Touvron et al., 2023). However, conventional wisdom suggests that scaling up model size does not improve utility in DP training since more model parameters will lead to lower signal-to-noise ratio[1]. To test this hypothesis, we train DP-Cap with different model sizes (Tiny, Small, Base) using the same hyper-parameters and evaluate their performance in Table 4; see Table 5 for details about different model sizes. We observe improvements when scaling up the model from DP-Cap-Tiny to DP-Cap-Small, as well as from Small to Base. Our observation suggests that DP-Cap has strong model scaling behavior even with DP-SGD training. This shed lights on further scaling up DP-Cap models to push the performance limit of differentially private vision-language foundation models for future work.

In Appendix B, we show additional results on Captioning (V-L) and on the impact of the compute budget on the performance of DP-Cap. We will open source our code.

## 5  DISCUSSION

We demonstrated that DP pre-training of vision-language foundation models is viable. In particular, image captioning is an ideal objective that supports both per-sample loss and large batch training — two critical ingredients in DP-SGD. When applied to the Dedup-LAION-233M dataset, the trained model learns useful image representations for downstream tasks and exhibits strong multimodal capabilities. Through our study we also identify three open problems in the general direction of DP pre-training of large-scale foundation models that are difficult to handle with existing techniques:

1. Scaling up the batch size to extreme levels is crucial for reducing the effective noise and facilitating model convergence. Is there a fundamental reason why image captioning can support extremely large batch training? When the model does not scale well with batch size, is it due to poor optimization or is it inherent to the model and/or task? Answers to these question can help identify other successful training recipes that unlock higher performance DP foundation models.

2. Scaling up the model size is typically beneficial in non-private foundation model training, but can be ineffective in DP training due to a decrease in gradient SNR. We have observed a performance improvement from Tiny to Base. However, we anticipate that deterioration could occur when scaling up to a much larger model size, especially when the signal-to-noise ratio (SNR) undergoes a certain threshold. Does this threshold exist? Can this trade-off be resolved through parameter-efficient architectures?

3. Contrastive learning offers unique advantages compared to other multimodal learning objectives such as strong zero-shot performance, but is not compatible with standard DP-SGD training. What techniques can enable differentially private contrastive learning?

---

[1]This is because the added noise has $L_2$ norm $\approx \sigma C \sqrt{d}/B$, where $d$ is the number of model parameters, whereas the gradient norm is constrained to $C$ regardless of model size.

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

# A  IMPLEMENTATION DETAILS

Table 5: Details of transformer backbone variants used in DP-CAP.

| Model | Encoder depth | Encoder width | Decoder depth | Decoder width | # parameters (encoder & decoder) |
|-------|--------------|--------------|--------------|--------------|--------------------------------|
| *DP-CAP-Tiny* | 12 | 384 | 6 | 384 | 59M |
| *DP-CAP-Small* | 12 | 576 | 6 | 576 | 115M |
| *DP-CAP-Base* | 12 | 768 | 6 | 768 | 190M |

## A.1  TRAINING DETAILS

**DP accounting.**  We use RDP accounting with subsampling from the Opacus library (Yousefpour et al., 2021). Let $D_\alpha$ denote the Rényi divergence of order $\alpha$ (Rényi, 1961), and let

$$g_\alpha(\sigma, q) := D_\alpha((1 - q)\mathcal{N}(0, \sigma^2) + q\mathcal{N}(1, \sigma^2) \,||\, \mathcal{N}(0, \sigma^2)). \tag{4}$$

Then, from Mironov et al. (2019), performing $S$ steps of DP-SGD satisfies $(\varepsilon, \delta)$-DP with:

$$\varepsilon := \min_\alpha \left\{ S \cdot g_\alpha(\sigma, q) + \frac{\log(1/\delta)}{\alpha - 1} \right\}. \tag{5}$$

The quantity $g_\alpha(\sigma, q)$ can be upper bounded mathematically or derived numerically: we use the Opacus (Yousefpour et al., 2021) library for accounting in our work.

Regarding the DP guarantee, $\varepsilon$-DP bounds the amount of information extracted from each training sample by $\varepsilon$. Notably, for DP-Cap, each sample is made of {image + caption}, while ViP utilizes {image} only. Consequently, DP-Cap inherently offers an equivalent or better privacy guarantee for each image. One way to see it is to note that DP provides protection against membership inference attacks (Shokri et al., 2017). Suppose $\varepsilon$-DP upper bounds the success rate of a membership inference attack (when given the image-text pair) against DP-Cap as $\leq p$. Then the MIA success rate when given only the image can be at most $p$ since the attacker has strictly less information. This is exactly the upper bound for the success rate of a membership inference attack against ViP. In other words, any attacker that can attack the {image+caption} models (such as DP-Cap) can also attack the image only models (such as ViP).

On the other hand, since the {image+caption} models utilize the caption, the privacy leakage from the text part of the image-caption pair is non-zero for $\varepsilon > 0$. It is worth noting that in our set up since we use DP, we protect the captions with the same $\varepsilon$-DP guarantee. Thus, the privacy protection for DP-Cap is neither strictly stronger nor strictly weaker than that for ViP, so the two privacy notions are not directly comparable.

**Model details and task description.**  We utilize a transformer architecture (Vaswani et al., 2017) DP-Cap. This captioner uses a text decoder that generates captions in an auto-regressive manner, utilizing a full attention mechanism on the vision encoder's output, as well as causal attention on the text. This architecture is closely aligned with the Cap architecture introduced in Tschannen et al. (2023). See Table 5 for details about the transformer architecture for different sizes. Except for comparison in Table 5, we present the results of our *base* models.

**Hyperparameters.**  Our choice of gradient clipping factor is $C = 1$, as we did not observe any performance improvement with other values. We always use AdamW (Loshchilov & Hutter, 2018) for training. We use a learning rate of $5.12 \times 10^{-4}$. The learning rate is kept constant across batch sizes for TAN simulations and for the performance comparison in Figure 3 as the effective noise is kept constant in these cases (Sander et al., 2023). We use a maximum length of 40 tokens to process the LAION captions. We use a linear schedule, with $40\%$ of warmup iterations, and $2\times$ the entire training as decay horizon. As opposed to what was previously observed (De et al., 2022; Sander et al., 2023), the learning rate schedule played an important role for us with DP-SGD training. We use a weight decay of $0.05$. These choices come from hyperparameter search using TAN simulation with our base model. Following the standard practice (Berrada et al., 2023; De et al., 2022; Li et al., 2021b; Yu et al., 2023; Sander et al., 2023), we do not count hyperparameter search within

Table 6: Different set-ups for ViP (Yu et al., 2023) pre-training: ImageNet-1k Linear probing.

| pretraining data | (B, $\sigma$, S) | ImageNet-1K | | | | |
| --- | --- | --- | --- | --- | --- | --- |
| | | 1-shot | 2-shot | 5-shot | 10-shot | full |
| LAION-233M | (97k, 0.48, 2400) | 2.5% | 4.1% | 8.5% | 14.2% | 55.7% |
| Dedup-LAION-233M | (97k, 0.48, 2400) | 2.5% | 4.2% | 8.5% | 14.3% | 55.8% |
| Dedup-LAION-233M | (1.3M, 0.73, 5708) | 2.7% | 4.6% | 9.4% | 15.7% | 56.5% |

our privacy budget. Liu & Talwar (2019) have shown that hyperparameter search might not incur observable privacy loss.

**Computation cost.** DP-SGD introduces additional computational overhead compared to non-private training. One reason is the computation of per-sample gradient norms: In our experiments, employing the ghost clipping technique (Li et al., 2021b) in conjunction with the Mixed Precision Package in PyTorch, each update computation was approximately two times slower than the non-private version. Crucially, to achieve a favorable privacy-utility trade-off, DP-SGD necessitates training with massive batches over a substantial number of steps to achieve a good privacy-utility trade-off, as elaborated in Section 3.2. All our hyperparameter search were performed using the TAN simulation for one epoch on our Dedup-LAION-233M. For our $\varepsilon = 8$ models, we limited training to 32 epochs, a process that took 6 days utilizing 128 V100 GPUs for the Base model. This exertion imposes a considerable energy consumption, resulting in elevated CO2 emissions. Our intention in releasing these models is to contribute to the mitigation of future carbon emissions, as the training has already been completed.

**Pre-training DP-Cap on the synthetic dataset.** Compared to the encoder and decoder architecture design used in masked autoencoders (MAE) (He et al., 2022b), the two main differences of the image captioning model used in this paper are: (1) The output of the encoder is fed into the decoder via cross-attention (Vaswani et al., 2017) in Cap; and (2) The self-attention used in the Cap decoder is causal self-attention. Similar to Yu et al. (2023), we apply the synthetic image dataset, Shaders21k (Baradad et al., 2022), to pre-train the DP-Cap model via MAE-based training. We follow most of the training setups used in ViP synthetic pre-training (Yu et al., 2023), except that we feed the output of the encoder to the decoder via cross-attention. The training loss of the synthetic pre-training in this paper is still the reconstruction loss used in MAE (He et al., 2022b), and we did not leverage real-world text data for pre-training. After the model is pre-trained on Shaders21k, we change the self-attention to causal self-attention in the decoder, and replace the final layer (for pixel-wise reconstruction) of the decoder with the (randomly initialized) decoding layer for next word prediction. After making these modifications, we apply DP-SGD to pre-train our DP-Cap model with standard image captioning training objectives (see Section 3.1).

**Pre-training ViP.** To conduct a comparison with training on an identical datasets, we follow the methodology outlined in (Yu et al., 2023) to train with DP-SGD a MAE-based model, but with a change in the training data from LAION-233M to Dedup-LAION-223M. We report the results using the corresponding ViP version in Figure 1 and Table 1. In Table 6, we further examine the linear probing performance on ImageNet and observe a marginal difference of only 0.01% between the original model and the one trained on the deduplicated dataset. In addition, to corroborate the observation made in Figure 3, which suggests that the MAE-based method struggles to effectively harness massive batch sizes for achieving low effective noise in DP-SGD, we also train a ViP model using the exact privacy parameters employed for DP-Cap (under $\varepsilon = 8$) and a notably large batch size of 1.3 million. For few-shot and full linear probing, we observe only a small improvement over the original ViP model that was trained with batch size 90k, but not for full linear probing. The success of DP-Cap is not solely attributed to its appropriate privacy parameters but is also a consequence of its remarkable ability to leverage the small effective noised induced by extremely large batch sizes.

Table 7: Compositional understanding (ARO): Results for CLIP (base) in Yuksekgonul et al. (2022) compared to our evaluation.

| Model | ARO | | | |
| --- | --- | --- | --- | --- |
| | VGR | VGA | COCO | Flickr |
| CLIP (eval from Yuksekgonul et al. (2022)) | 59% | 63% | 46% | 60% |
| CLIP (our eval) | 62.4% | 62.9% | 47.8% | 58.0% |

## A.2 EVALUATION DETAILS

### A.2.1 DETAILS ABOUT ZERO-SHOT IMAGE CLASSIFICATION

While methods employing contrastive learning, such as CLIP, excel in this task, captioning methods exhibit comparatively lower performance, and with greater computational demands during evaluation. To evaluate a captioning model's zero-shot performance, we employ two distinct strategies:

- **Tree-based search**: We initiate caption generation with a prompt like "this is a photo of a," and greedily select the most likely next token among those that lead to valid completions within the true label set. The process continues until an End of Sentence (EOS) token is reached. For instance, if there are only two labels starting with "car": "car [EOS]" and "carpet [EOS]", and the initial predicted token is "car". Then the text decoder will predict the next token among "[EOS]" and "pet". If, among these two, "[EOS]" is chosen, and "car [EOS]" corresponds to the true label, then the zero-shot prediction is deemed correct.
- **Loss-based classification**: We assess, for each image, the probability of various captions that begin with "this is a photo of a [...]" where "[...]" is substituted with all feasible labels. Subsequently, we select the label that yields the most probable caption.

The "loss-based classification" comes with significantly higher computation costs as all the different captions have to be evaluated for each image (there representations is conditional to the image). For ImageNet, it implies 1000 forwards through the decoder for each image. We thus employ the tree-based search for presenting our findings in Table 2, although its greedy character with no backtracking is not optimal. Surprisingly, our preliminary experiments suggest the tree-based search gives comparable results.

### A.2.2 DETAILS ABOUT ARO EVALUATION

We adhered to the protocol and codebase established in Yuksekgonul et al. (2022) for re-evaluating CLIP's performance, and we observe slightly different results (see Table 7). For our captioning models, our approach involved computing the cross-entropy loss for all possible captions associated with each image and subsequently selecting the one with the lowest loss.

### A.2.3 DETAILS ABOUT LINEAR PROBING AND FINETUNING EVALUATION.

Few-shot linear probing is accomplished using the Cyanure library (Mairal, 2019). We use the same hyper parameters as in Assran et al. (2022). We adapted the MAE (He et al., 2022b) codebase for full linear probing, and we use the same hyperparameters as in Yu et al. (2023) (extract 12 layers of the image encoder, LARS optimizer (You et al., 2017) with base learning rate of 0.1, no weight decay and batch size of 16384).

## B ADDITIONAL RESULTS

### B.1 ADDITIONAL EXPERIMENTS

**Impact of the initialization (V)** Our synthetic initialization for DP-Cap achieves less favorable results than the one from ViP reaches 50% (Yu et al., 2023); for instance, for full linear probing

Table 8: Training from random initialization: Superiority of DP-Cap over ViP, both trained from random initialization.

| Model | ImageNet-1K | | | |
|---|---|---|---|---|
| | 1-shot | 2-shot | 10-shot | full |
| ViP ($\varepsilon = 8$) | 0.1% | 1.7% | 6.1% | 23.9% |
| DP-Cap ($\varepsilon = 8$) | 5.6% | 8.5% | 18.8% | 47.0% |

Table 9: Fine-tuning evaluation on few-shot downstream classification.

| Model | Aircraft | | | Caltech-101 | | | CIFAR-100 | | |
|---|---|---|---|---|---|---|---|---|---|
| | 10-shot | 20-shot | 30-shot | 5-shot | 10-shot | 30-shot | 5-shot | 10-shot | 30-shot |
| AlexNet | 23.3% | 34.4% | 41.4% | 64.7% | 73.6% | 81.4% | 29.7% | 36.3% | 49.3% |
| SimCLR | 38.8% | 56.9% | 64.9% | 81.7% | 89.1% | 94.5% | 49.9% | 60.2% | 71.8% |
| TAN | 22.8% | 37.9% | 46.0% | 49.3% | 66.4% | 77.9% | 21.3% | 27.8% | 42.4% |
| ViP | 31.6% | 53.1% | 64.3% | 68.1% | 79.0% | 88.9% | 30.7% | 41.0% | 57.5% |
| DP-Cap | 37.5% | 57.9% | 66.7% | 70.3% | 81.3% | 90.0% | 36.3% | 46.3% | 62.1% |

on ImageNet, they achieve 44% (Figure 2) and 50% respectively. However we have demonstrated that training with DP on top of synthetic initialization leads to significantly better results for DP-Cap compared to ViP for all the metrics; see Table 1, Table 9 and Figure 1. We observe that this superiority also appears when the models are trained from random initialization: as shown in Table 8, the improvement over ViP is even larger when training without synthetic initialization.

**Fine-tuning (V).** In Table 9, we present DP-Cap's performance in fine-tuning for few-shot evaluation. In contrast to the linear probing results shown in Table 1, the network is completely unfrozen. Therefore, we assess DP-Cap's capabilities primarily as a network initialization. Similarly to the linear probing results, we note a significant improvement in all metrics compared to previous DP vision backbones. Note that, similarly to linear probing comparison in Figure 1, we compare to non-private model performance which provides information about the performance gap between private models and non-private models.

**Captioning task (V-L).** We evaluate the image captioning performance of DP-Cap in comparison to non-private Cap. In Fig. 1(c), we present some (randomly chosen) captions generated by DP-Cap; more examples for DP-Cap and Cap can be found in Appendix B.3. Qualitatively, DP-Cap seems to generate reasonably good captions, similar to the ones generated by Cap. We also compare the two models quantitatively using the CIDEr metric (Vedantam et al., 2015) to evaluate the generated captions on the MS-COCO test set, and the results are summarized in the last column of Table 2. As DP-Cap and Cap are only trained on noisy captions from LAION, the CIDEr metric on MS-COCO is relatively low for both models. Moreover, despite the similar performance between DP-Cap and Cap on ARO, the gap is much more significant for the captioning evaluation. Given these results, it is plausible that even though DP-Cap attains remarkably compositional understanding capabilities, its ability to generate text is still limited.

We also finetune Cap and DP-Cap's decoders (while freezing the encoder) for one epoch on the MS-COCO train set, and assess the improvement in CIDEr scores in Table 10 to showcase the quality of the image representations and decoder initialization from the pre-training stage. The captions in Figure 1 and Appendix B.3 are generated using models that were *not* trained on MS-COCO.

**What can we do with more compute budget?** We restricted training the DP-Cap model for a compute budget of 32 epochs on the Dedup-LAION-233M dataset for each of our models with $\varepsilon = 8$. To fit the privacy budget while utilizing a batch size of 1.3 million and training for 32 epochs, RDP analysis yields $\sigma = 0.728$. However, we anticipate that further increasing the compute budget can yield even better models up to a certain limit: With the same $\varepsilon$ and batch size, doubling

Table 10: Captioning evaluation on the MS-COCO test set of Cap and DP-Cap. For "finetuned", the models's decoders are finetuned for one epoch on the MS-COCO train set (with the image encoder frozen).

| Model | CIDEr score | |
| | original | finetuned |
| --- | --- | --- |
| Cap | 29.9 | 79.2 |
| DP-Cap | 15.7 | 51.3 |

Table 11: TAN simulation of the impact of the compute budget on the performance at fixed $B$.

| | $\sigma$ | |
| | 0.81 | 0.95 |
| --- | --- | --- |
| Epochs | 64 ($\times 2$) | 128 ($\times 4$) |
| Effective noise $\sigma/B$ | $\times 1.12$ | $\times 1.32$ |
| Predicted Final loss | $-0.2$ | $-0.2$ |
| Predicted 10-shot ImageNet | $+5\%$ | $+5\%$ |

the compute to 64 epochs only necessitates a 12% increase in $\sigma$. This increase enables twice as many steps to be performed with only a marginal increase in effective noise, potentially allowing the model to converge to a better solution.

In the absence of necessary compute for running this experiment, *we partially validate this hypothesis through the Total Amount of Noise (TAN) simulation*, training for the same number of gradient steps and with the same SNR per step, but using a $\times 32$ smaller batch size and $\times 32$ smaller $\sigma$ to simulate at $\times 32$ lower compute. Our results in Table 11 indicate a significant performance improvement of 5% in 10-shot accuracy on ImageNet (compared to a similar simulation of the 32 epochs training). However, increasing the budget further to 128 epochs does not seem to enhance performance compared to the 64 epoch counterpart. Intuitively, the lower gradient SNR and larger number of gradient steps have opposite effects on optimization, and pushing past the "sweet spot" of training for 64 epochs at $\sigma = 0.81$ results in noisy steps that are unproductive for model convergence. To surpass the performance of the 64-epoch, 1.3-million batch size DP-Cap model, training with an even larger batch size appears necessary. We emphasize again that this result is derived through TAN simulation, and actual, compute-intensive training is required to fully validate this assertion.

### B.2 MORE ON THE IMPACT OF DATASET SIZE AND PRIVACY PARAMETERS

**Dataset size.** We emphasize here (again) the importance of having enough training data to achieve a good privacy-utility trade-off with DP-SGD. As depicted in Figure 4, increasing the number of training samples $N$ while keeping the same number of equivalent DP-SGD steps (*i.e.*, keeping batch size $B$, noise $\sigma$, and number of update steps $S$ constant) considerably reduces the privacy budget $\varepsilon$. Equivalently, having more data allows for an increase in the number of equivalent DP-SGD steps at fixed $\varepsilon$. Similar observations were also made by Tramer & Boneh (2020); McMahan et al. (2017). The abundance of pre-training data available for training foundation models thus proves highly compatible with DP requirements.

**Batch size and $\sigma$.** We wish to underscore the influence of batch size and $\sigma$ on both the computational budget and model performance. As highlighted in Section 3.2, for a given target $\varepsilon$, elevating $\sigma$ beyond 0.5 allows training for significantly more steps. In Figure 5, the blue, orange and green lines show the batch size ($B$) vs. compute trade-off ($E$) at a given $\sigma$. The lines are monotonically decreasing with $B$, signifying that the number of epochs $E$ decreases when increasing $B$. When maintaining a fixed privacy budget $\varepsilon = 8$, even a marginal increase in $\sigma$ from 0.48 to 0.728 (from blue to orange) translates to a remarkable increase ranging from 100 (for small batch sizes) to 100,000 (for very large batch sizes) times more gradient steps. Thus it is favorable to increase $\sigma$ and $B$ at the same time for better model convergence.

Meanwhile, doing so also incurs a higher computational cost: Under a 32-epoch budget on Dedup-LAION-233M with a batch size of 1.3 million, we had to cut the red curve in Figure 5, with

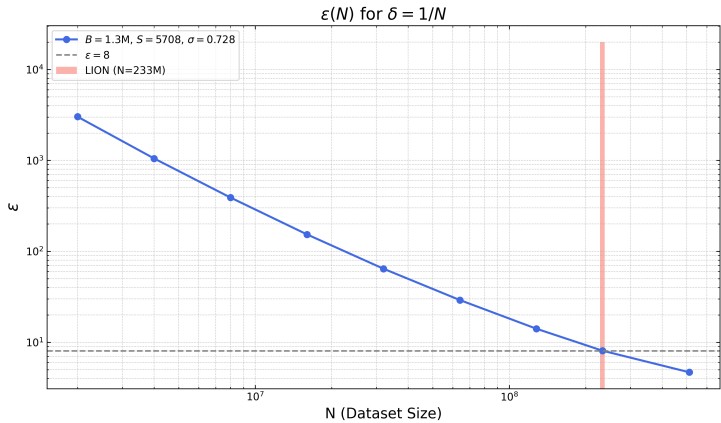

Figure 4: At fixed (B, $\sigma$, S), $\varepsilon$ drastically reduces with the dataset size.

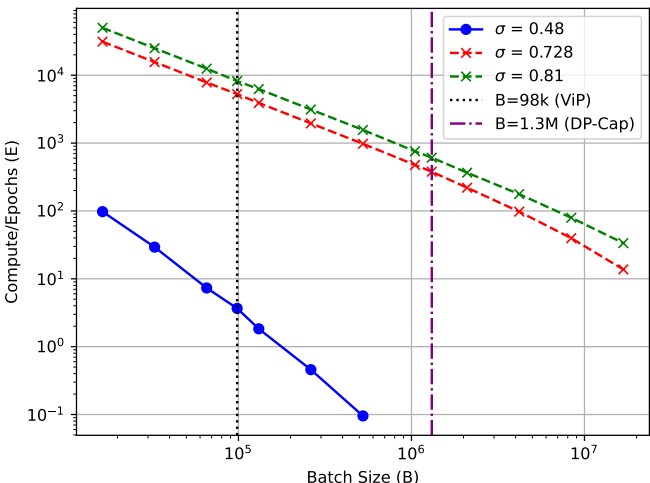

Figure 5: All points correspond to $\varepsilon = 8$ for a dataset of size $N = 233$M. At fixed $\varepsilon$ and $\sigma$, the number of epochs decreases as the batch size increases.

$\sigma = 0.728$. As outlined in Section 4.3, with twice this budget, we could have raised $\sigma$ to 0.81 (green curve), with simulations indicating that this would have substantially improved performance. Additionally, Section 3.2 underscores that increasing the batch size is pivotal for achieving a high SNR while maintaining reasonable privacy guarantees. It is also crucial to note that at fixed $\varepsilon$, the compute budget is inversely proportional to the batch size. Therefore, increasing the batch size is beneficial for both SNR and computational efficiency. However, an excessively large batch size leads to fewer epochs and consequently a very limited number of training steps, which is detrimental to the training process (in addition to the difficulties of large batch training). For optimal privacy-utility-compute trade-off, a balance must be struck between computational resources, feasible batch size, and a reasonable number of training steps.

### B.3  IMAGE CAPTION EXAMPLES

In Figures 6 and 7, we show images from the MS-COCO 2017 test set and their corresponding captions generated by human annotator, Cap, and DP-Cap. Images in Figure 6 are selected randomly, whereas images in Figure 7 are randomly selected from the top 10% CIDEr score examples for DP-Cap. Qualitatively, the human-generated captions are more precise, whereas the captions generated

by Cap and DP-Cap are more generic and sometimes contain factual errors. This is to be expected since Cap and DP-Cap are trained on LAION with much noisier text description and were *not* fine-tuned on MS-COCO. Nevertheless, DP-Cap still generates grammatically correct and (mostly) semantically coherent captions for unseen images.

True: A brown vase has four black horses on it.
DP-CAP: vintage japanese hand painted lacquer box
CAP: a greek vase depicting the battle of hastings, c.

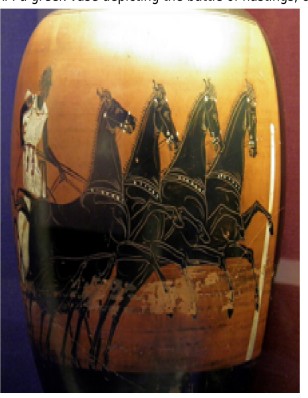

True: a pack of elephants grazing in a dirt enclosed space
DP-CAP: elephants are seen in the zoo in the city of london, england, on july 1, 2020.
CAP: parkville, victoria - australia'melbourne zoo - d

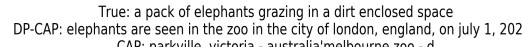
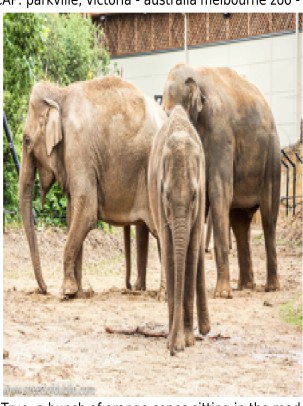

True: A pastry is torn in half on a plate.
DP-CAP: how to make a cake with a cake mix
CAP: baked potato with cheese and honey

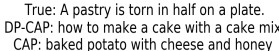
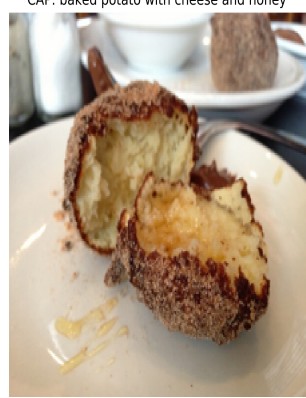

True: a bunch of zebras out in a grassy field
DP-CAP: zebras grazing in the field
CAP: zebras and wildebeests grazing in the serengeti n

True: a bunch of orange cones sitting in the road
DP-CAP: the new york city fire department's fire department responded to a fire at the scene of a fi re at the scene of a fire at the intersection of broadway and broadway in new york on tuesday
CAP: construction continues on the new downtown east s

True: A cat is on a desk looking at the computer screens.
DP-CAP: how to make a home theater
CAP: cat sitting at desk with computer

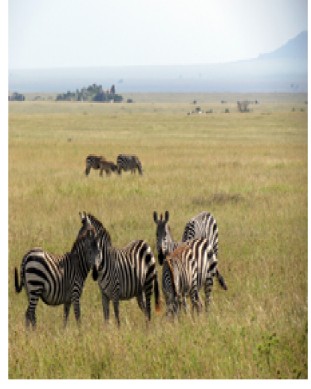
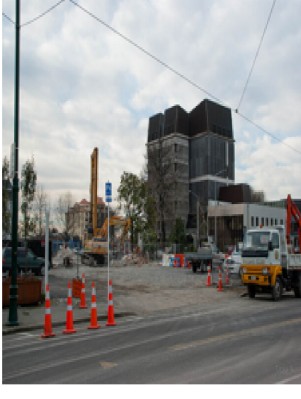
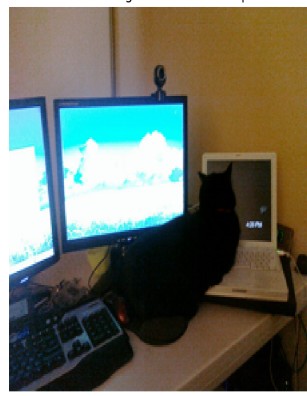

True: Bathroom with orange shower and mirror/cabinet curtains
DP-CAP: how to make a shower curtain
CAP: how to make a shower curtain

True: two stuffed animals posed together in black and white
DP-CAP: " " " " " " " " i love you " " " " - - - - - - - - - - - - - - - - - - - - - -
CAP: stuffed animals

True: A plate of food sitting on a table, it appears to be a salad.
DP-CAP: grilled chicken with cilantro lime dressing
CAP: roasted broccoli and lentils with black rice and

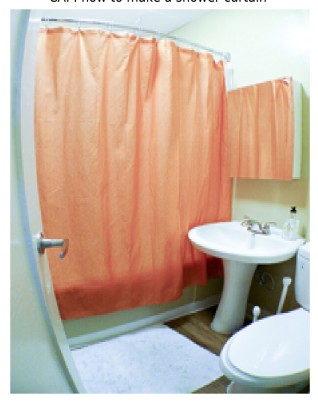
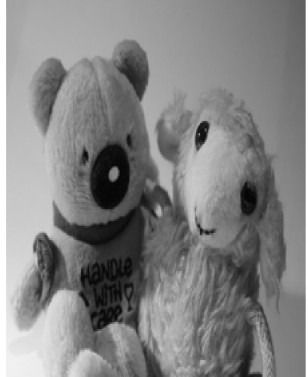

Figure 6: Captions of randomly selected images from the MS-COCO 2017 test set.

True: a bird feeder that is attached to a tree
DP-CAP: a small bird feeder made from a plastic bottle
CAP: how to attract birds to your yard

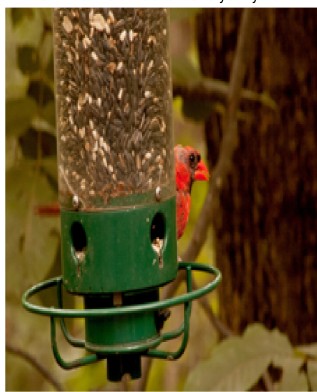

True: A living room with a white sofa and a gray rug.
DP-CAP: living room with a large sofa and a coffee table.
CAP: living room with carpet, hardwood floors, and a d

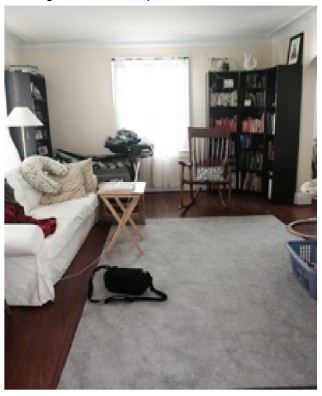

True: A dog is running on the beach sand.
DP-CAP: white horse running on the beach
CAP: white dog running on the beach

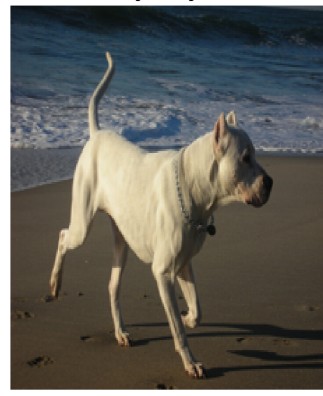

True: White swans swimming in a harbor with docked boats.
DP-CAP: swans swimming in the water
CAP: swans swimming in the harbor

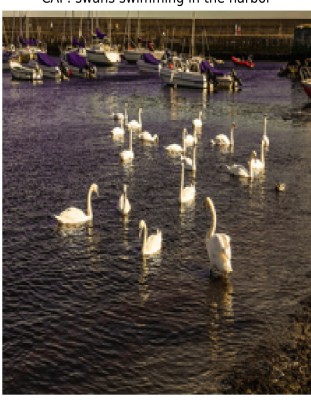

True: Two large brown elephants walking in a grassy field.
DP-CAP: a herd of elephants walking in the grass
CAP: elephants walking in the park

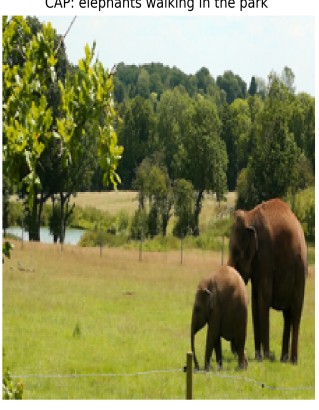

True: A tan dog eating food scraps from a plate.
DP-CAP: dog eating food
CAP: dog eating

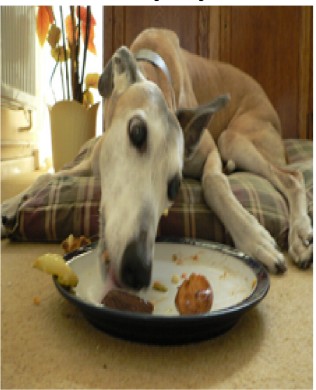

True: A seagull standing near the ocean on the sand.
DP-CAP: seagull standing on the beach
CAP: seagull standing on the beach

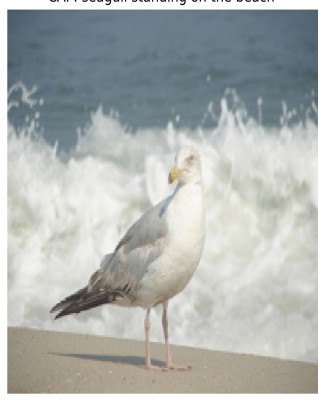

True: A polar bear swimming underwater, approaching some rocks.
DP-CAP: a polar bear swimming in the water
CAP: swimming polar bear

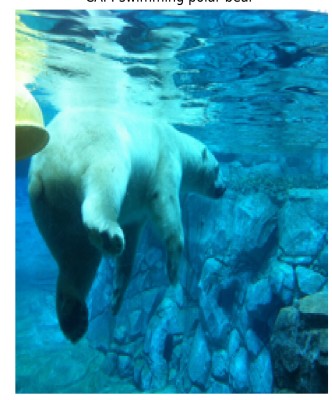

True: a cat wearing a hat on its head
DP-CAP: cat wearing a pink hat
CAP: cat wearing pink hat

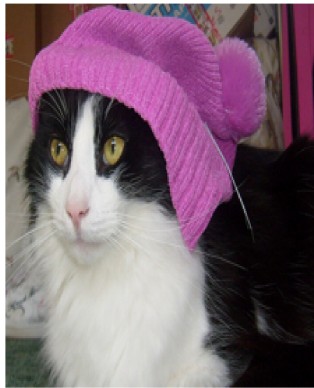

Figure 7: Captions of images (randomly picked among the top 10% CIDEr score of DP-Cap) from the MS-COCO 2017 test set.

