# OpenReview forum: "Differentially Private Vision-Language Foundation Models via Image Captioning"
_ICLR.cc/2024/Conference — Submitted to ICLR 2024_

### Official Review · Reviewer_aDje · 2023-10-30

**Soundness:** 4 excellent
**Presentation:** 4 excellent
**Contribution:** 2 fair
**Rating:** 6
**Confidence:** 4

**Summary:**

The paper presents a novel approach for learning differentially private vision-foundation models via image captioning. The key innovation of the method is the use of an image captioning objective that enables the direct application of DP-SGD, since the loss is an additive function over the samples. Compared to the existing approach of using a masked autoencoder objective, the proposed image caption objective not only enables vision-language tasks, but allows for the use of much larger batch sizes during model training, without degrading the model performance, which greatly reduces the effective noise per training iteration, leading to better privacy-utility tradeoffs. In experimental evaluations on downstream vision and vision-language tasks, the method achieves SoTA performance relative to DP baselines and, on some tasks, even comparable performance to non-private models.

**Strengths:**

**Clarity**
* The writing, figures, and tables are very clear.

**Significance (Very Strong results)**
* On the linear probing classification, the proposed method performs comparably or better than non-private AlexNet and SimCLR, and meaningfully outperforms the SoTA DP baseline ViP.
* On few-shot learning, the proposed method more than doubles the accuracy of the SoTA baseline ViP.
* On ARO, the proposed method performs comparably or better than non-private CLIP.

**Weaknesses:**

**Limited Novetly**
* The paper lacks any meaningful theoretical or algorithmic contributions. The only insight of the paper is that image captioning is a suitable loss for DP training.

**Questions:**

* What is the reason for the *dilemma of large batch training in DP-SGD*, and how does image captioning solve this dilemma? Can you provide any insights?

* Would combining the MAE and image captioning objectives lead to even better performance?

---

> ### Author Response · Authors · 2023-11-18
> **Rebuttal by Authors**
>
> We thank the reviewer for their insightful comments and questions.
>
> >**Q1**: *Limited Novetly The paper lacks any meaningful theoretical or algorithmic contributions. The only insight of the paper is that image captioning is a suitable loss for DP training.*
>
> **A1**: The vast majority of existing work on DP training operates on a single modality, be it image classification, image-only SSL, text classification, language modeling, etc. Our work is the first to push beyond this barrier and show that DP can also be successfully applied to the currently mainstream paradigm of multimodal learning. We intentionally stuck to well-established and proven techniques for this purpose, using the standard DP-SGD algorithm and Renyi privacy accounting, and the Cap algorithm for image captioning pre-training. By doing so, we hope to establish a strong baseline for DP pre-training on multimodal data, and lay the groundwork for future research by releasing our code/model so that others can build upon our successful training recipe.
>
> Nevertheless, our work does reveal several novel and interesting findings regarding the suitability of image captioning for large-scale DP training, including its ability to scale to massive batch sizes as large as 1.3M, and its synergy with the TAN simulation framework for efficient exploration of training algorithms and hyperparameters. These findings, although not novel in terms of techniques, are crucial for future research since they collectively form a comprehensive strategy for scaling up DP training.
>
> >**Q2**: *What is the reason for the dilemma of large batch training in DP-SGD, and how does image captioning solve this dilemma? Can you provide any insights?*
>
> **A2**: The observation that training with massive batches can deteriorate the performance with DP-SGD [SSS2023] is also present in the non private literature, e.g. with SGD. DP-Cap exhibits robust scaling capabilities concerning batch size, contributing to its success. Our intuition is that a captioner training resembles a language model training, which can usually deal with larger batches. We believe that the same observation could be made for classical (non-private) SGD.
>
> >**Q3**: *Would combining the MAE and image captioning objectives lead to even better performance?*
>
> **A3**: We thank the reviewer for this interesting observation.  It would mean performing backpropagation on the sum of these two losses, from image reconstruction and captioning. Intuitively, we would expect that this joint optimization may not be inherently advantageous, given that image representations learned from reconstruction and captioning are not necessarily similar. It is also an interesting question outside of the scope of DP that is worth exploring.
>
> We hope that we have adequately addressed the queries posed by the reviewer. We are readily available for any further discussions or clarifications as needed. Please note that we have also added a general response that details the changes in the revised manuscript.
>
> [SSS2023] TAN Without a Burn: Scaling Laws of DP-SGD. Tom Sander, Pierre Stock, Alexandre Sablayrolles. ICML 2023.

---

### Official Review · Reviewer_nthH · 2023-10-31

**Soundness:** 3 good
**Presentation:** 3 good
**Contribution:** 3 good
**Rating:** 6
**Confidence:** 3

**Summary:**

This paper focuses on training data-safe foundation model under the differential privacy framework. Existing DP-based foundation models have difficulty in learning effective representations. To address this problem, the authors propose to take advantage of the large amount of paired image-text data to train multimodal foundation models. Specifically, the proposed model is trained to generate image captions under the DP-framework. Experiments on several image recognition benchmarks as well as some ARO benchmarks validate the effectiveness of the proposed method.

**Strengths:**

(1) This paper is well-written. The technical details are clearly presented and it is easy for the readers to follow the ideas of the proposed method.
(2) The proposed method is effective as shown in the experiments. Specifically, the proposed DP-cap model significantly surpasses several existing DP-based methods and is on par with AlexNet and CLIP in some specific scenarios.
(3) The authors have described some interesting findings in their experiments, e.g. the impact of batch size, which is different from the behavior of some existing methods (e.g. ViP). Such phenomenon might be inspiring for future researchers.

**Weaknesses:**

(1) The technical novelty of the proposed method is somehow limited. To be specific, the proposed method consists of two training stages. The first stage trains the model on texture images similarly as proposed in Yu er al. (2023). Next, the second stage further trains the model on image captioning task under the DP framework, which is quite straightforward.
(2) I am not an expert in model safety, yet I am wondering that except for DP, is there any other frameworks/methods for overcoming the copyright/privacy issue? It is clear from the manuscript that DP-based models performs much inferior to non-DP-based models, yet it is not clear is there any alternative to DP. If so, why does the authors choose DP-based method at first place?

**Questions:**

(1) As shown in Appendix A.1, the authors used 128 V100 GPUs for training the model. As a result of the huge amount of demanded GPUs, it would be difficult for most of the researchers in this area to follow this work. On the other hand, as shown in Figure 3(b), the proposed method has similar loss values across different batch sizes. Therefore, it would be interesting to see if the proposed method can achieve similar performances with smaller batch size. If so, more researchers would be able to follow and further improve this work.
(2) As discussed in the “weakness” part, is there any other promising framework/method that could overcome the copyright/privacy issues of large foundation models? If so, why do the authors choose DP-based methods in this manuscript?
(3) I would expect the authors to further clarify the technical novelty of the proposed method.

---

> ### Author Response · Authors · 2023-11-18
> **Rebuttal by Authors (Part 1)**
>
> We thank the reviewer for their insightful comments and questions.
>
> >**Q1**:*“The technical novelty of the proposed method is somehow limited. To be specific, the proposed method consists of two training stages. The first stage trains the model on texture images similarly as proposed in Yu er al. (2023). Next, the second stage further trains the model on image captioning task under the DP framework, which is quite straightforward.”*
>
> **A1**:  The majority of prior work on Differentially Private training has been confined to a single modality, such as image or text classification, language modeling, or image-only Semi-Supervised Learning. Our research breaks through this limitation, marking the first successful application of DP for multimodal learning. Deliberately relying on established techniques, like the standard DP-SGD algorithm with Renyi DP accounting and the Cap algorithm for image captioning pre-training, we aim to set a robust benchmark for DP pre-training on multimodal data. By openly sharing our code and model, our intention is to enable others to build upon this effective training approach and propel future research in this direction.
>
> Our study also uncovered significant and original insights regarding the viability of image captioning in large-scale DP training. Notably, its capability to scale to massive batch sizes, reaching up to 1.3 million, and its synergistic relationship with the TAN simulation framework for efficient exploration of training algorithms and hyperparameters. While these insights might not introduce new techniques, they collectively form a fundamental strategy for expanding DP training at scale, offering valuable directions for future research.
>
> **Q2**: *“I am not an expert in model safety, yet I am wondering that except for DP, is there any other frameworks/methods for overcoming the copyright/privacy issue? It is clear from the manuscript that DP-based models performs much inferior to non-DP-based models, yet it is not clear is there any alternative to DP. If so, why does the authors choose DP-based method at first place?”*
>
> **A2**: This is a great question. As the topic of copyright/privacy is inherently multidisciplinary, the choice of solution must be acceptable not only to the research community, but also in the eyes of law practitioners and regulators. DP stands out in this regard as the only existing solution that is not only provably rigorous and thus scientifically appealing, but also enjoyed adoption as the gold standard for privacy protection by organizations such as the US Census Bureau, Google, Apple, etc. [Des21, CDE+2023]. The goal of our paper is to push the boundary of DP foundation model training and bring this gold standard notion closer to current practice in ML.
>
> We would like to emphasize that even though DP training indeed performs inferior to non-DP training, the latter may be prohibited by law or policy when applied to private data. Thus, it is more appropriate to view the non-DP result as the maximum attainable model performance rather than as a baseline. In practice, however, a portion of the training data may be publicly accessible. In such situations, a natural way to apply our findings is to initialize the model by non-DP training on the public data, then fine-tune with DP on the private data.
>
> [Des21] Damien Desfontaines. A list of real-world uses of differential privacy. https://desfontain.es/privacy/real-world-differential-privacy.html, 10 2021. Ted is writing things (personal blog).
>
> [CDE+2023] Rachel Cummings, Damien Desfontaines, David Evans, et al. Challenges towards the Next Frontier in Privacy. arXiv preprint arXiv:2304.06929.

---

> ### Author Response · Authors · 2023-11-18
> **Rebuttal by Authors (Part 2)**
>
> >**Q3**: *As shown in Appendix A.1, the authors used 128 V100 GPUs for training the model. As a result of the huge amount of demanded GPUs, it would be difficult for most of the researchers in this area to follow this work. On the other hand, as shown in Figure 3(b), the proposed method has similar loss values across different batch sizes. Therefore, it would be interesting to see if the proposed method can achieve similar performances with smaller batch size. If so, more researchers would be able to follow and further improve this work.*
>
> **A3**: Differentially Private training indeed places a significant computational burden, and our aspiration is that the foundational models trained with DP can be a one-time investment. In our work, we used the strategy from [SSS2023] to simulate training with smaller batches to alleviate the computational cost associated with HP search. The curves we plot in Figure 3.b. for smaller batches serve as a basis for this simulation, but do not have reasonable strict privacy guarantees. However, in settings where the simulation can closely match the result obtained with large batch training, it may be possible to conduct exploratory research entirely via simulation using much less compute.
>
> >**Q4**: *As discussed in the “weakness” part, is there any other promising framework/method that could overcome the copyright/privacy issues of large foundation models? If so, why do the authors choose DP-based methods in this manuscript?*
>
> **A4**: See answer to **Q2**. There are some alternative privacy-preserving ML notions  but they are not as rigorous and widely accepted as DP. We fully agree with the reviewer that DP has several caveats, but hope that our paper shows promise for its application in large scale settings.
>
> >**Q5**: *I would expect the authors to further clarify the technical novelty of the proposed method.*
>
> **A5**: See answer to **Q1**.
>
> We hope that we have adequately addressed the queries posed by the reviewer. We are readily available for any further discussions or clarifications as needed. Please note that we have also added a general response that details the changes in the revised manuscript.
>
>
> [SSS2023] TAN Without a Burn: Scaling Laws of DP-SGD. Tom Sander, Pierre Stock, Alexandre Sablayrolles. ICML 2023.

---

> > ### Comment · Reviewer_nthH · 2023-11-21
> > **Discussion**
> >
> > Most of my concerns are addressed. However, I still believe that the novelty of the paper is limited. Indeed, the paper gives some insight about training DP-based models with image captioning task. However, the paper is more like a technical report rather than an academic paper.

---

> > > ### Author Response · Authors · 2023-11-21
> > >
> > > We thank the reviewer for their response.
> > >
> > > We strongly believe that the merit of a paper should also be evaluated based on how much it contributes to the existing literature, rather than solely on its technical novelty. In line with this principle, we believe our article makes a noteworthy scientific contribution by showing that DP training with captioning is more sample-efficient than with reconstruction-based SSL. It's essential to highlight that our improvements are substantial and represent a significant leap beyond prior endeavours. We will also release our model and code so that our work can benefit the entire research community and help push forward the limits of private learning, which we believe is one of the fundamental objectives of academic research.

---

> > > > ### Comment · Reviewer_nthH · 2023-11-22
> > > >
> > > > Please note that I have always agreed that this paper has its value in some aspects, and I am not denying the contribution of this paper. However, as a paper submitted to a top-tier academic conference, I would expect more novel insights or methods in this paper. Therefore I gave a rating of 6 in my review.
> > > >
> > > > I sincerely suggest the authors to improve the paper in terms of novelty and depth in analysis if it got rejected.

---

### Official Review · Reviewer_NLQ1 · 2023-11-01

**Soundness:** 3 good
**Presentation:** 2 fair
**Contribution:** 2 fair
**Rating:** 6
**Confidence:** 3

**Summary:**

This paper demonstrates the feasibility of differentially private (DP) pre-training for vision-language foundation models. Specifically, it illustrates that DP pre-training, when applied to both images and captions, learns more information for downstream tasks, compared to pre-training on images alone. Additionally, the paper introduces a model named DP-Cap, which outperforms other existing models across several benchmarks.  A comprehensive ablation study is conducted in experiments, which shows the impact of different components on DP-Cap. Furthermore, the paper provides discussions regarding the challenges, potential directions, and open questions of DP pre-training large-scale foundation models.

**Strengths:**

+ This paper studies an important and interesting topic: security and privacy risks of foundation models.

It proposes training foundation models by combining differential privacy with vision-language pre-training. This approach can help high-utility foundation models with DP training on both images and captions compared with training on images only.

+ The paper introduces the DP-Cap model, which uses text supervision to learn image representations and exhibits multi-modal capabilities for DP training. The model achieves better performance on several benchmarks than several existing models.

+ A lot of interesting and useful discussions are provided in Section 3.1 and Section 5. They can provide insights into DP pre-training of vision-language foundation models and may provide some potential research directions.

+ The related work is well summarized. It can provide readers with a better understanding of the challenges and research status of this topic.

**Weaknesses:**

- There may be fairness issues in experimental comparisons.

    For example, the paper states that DP-Cap and ViP were trained using the same training dataset and privacy budget. However, it is worth noting that ViP was trained without utilizing captions (If I understand correctly), while DP-Cap leveraged both images and captions. Consequently, the observed performance improvement of DP-Cap compared to ViP could be attributed to the additional information used during training. While this improved performance may be seen as a contribution of DP-Cap, it is important to consider that the privacy risk might have increased since the model learned caption information.


- The effectiveness of the proposed method appears to lack a thorough analysis.

   In Figure 1(a), the accuracies of DP-Cap are reported as 0.58 and 0.33 under 'random init' and 'syn init' on ImageNet. It is evident that 'syn init' significantly contributes to DP-Cap's performance improvement. However, it's worth considering that 'syn init' is derived from the paper proposing ViP. An important question arises: if both ViP and DP-Cap were not to utilize 'syn init,' what impact would it have on their final results?"

- Typos. For example

  - The citation formats for some related papers appear to be incorrect.
  - Is the double line in Table 1 positioned correctly?"

**Questions:**

I am more curious whether using caption pre-training models increases privacy risks because the model sees more information.

---

> ### Author Response · Authors · 2023-11-18
> **Rebuttal by Authors**
>
> We thank the reviewer for their insightful comments and questions.
>
> >**Q1**: *“There may be fairness issues in experimental comparisons. For example, the paper states that DP-Cap and ViP were trained using the same training dataset and privacy budget. However, it is worth noting that ViP was trained without utilizing captions (If I understand correctly), while DP-Cap leveraged both images and captions. Consequently, the observed performance improvement of DP-Cap compared to ViP could be attributed to the additional information used during training. While this improved performance may be seen as a contribution of DP-Cap, it is important to consider that the privacy risk might have increased since the model learned caption information.”*
>
> **A1**: $\varepsilon$-DP bounds the amount of information extracted from each training sample by $\varepsilon$. Notably, for DP-Cap, each sample is made of {image + caption}, while ViP utilizes {image} only. Consequently, DP-Cap inherently offers an equivalent or better privacy guarantee for each image. One way to see it is to note that DP provides protection against membership inference attacks [SSS+2017]. Suppose $\varepsilon$-DP upper bounds the success rate of a membership inference attack (when given the image-text pair) against DP-Cap as $\leq p$. Then the MIA success rate when given only the image can be at most $p$ since the attacker has strictly less information. This is exactly the upper bound for the success rate of a membership inference attack against ViP. In other words, any attacker that can attack the {image+caption} models (such as DP-Cap) can also attack the image only models (such as ViP).
>
> On the other hand, since the {image+caption} models utilize the caption, the privacy leakage from the text part of the image-caption pair is non-zero for $\varepsilon>0$. It is worth noting that in our set up since we use DP, we protect the captions with the same $\varepsilon$-DP guarantee. Thus, the privacy protection for DP-Cap is neither strictly stronger nor strictly weaker than that for ViP, so the two privacy notions are not directly comparable.
>
> We thank the reviewer for this remark, and are adding this precision in the revised manuscript (Section A.1).
>
> >**Q2**: *“The effectiveness of the proposed method appears to lack a thorough analysis. In Figure 1(a), the accuracies of DP-Cap are reported as 0.58 and 0.33 under 'random init' and 'syn init' on ImageNet. It is evident that 'syn init' significantly contributes to DP-Cap's performance improvement. However, it's worth considering that 'syn init' is derived from the paper proposing ViP. An important question arises: if both ViP and DP-Cap were not to utilize 'syn init,' what impact would it have on their final results?"”*
>
> **A2**: We have observed that training with DP on top of synthetic initialization leads to significantly better results for DP-Cap compared to ViP. To answer the reviewer’s question, this superiority also appears when the models are trained from scratch: the improvement over ViP is even larger when training without syn-init. For instance, when trained from random initialization, ViP reaches 23.9% for full linear probing, while we achieve 47.0%. It naturally underscores the fact that our captioning approach is significantly better for differentially private representation learning for vision data. We acknowledge the significance of this observation. We have incorporated it into the updated paper for clarity, and added more numbers for comparison in Table 8 (Section B).
>
> >**Q3**: *Typos. For example: The citation formats for some related papers appear to be incorrect. Is the double line in Table 1 positioned correctly?"*
>
> **A3**: We thank the reviewer for these typos; we are happy to correct the citations. Could you please clarify which specific related work you are referring to? For the double line, we wanted to highlight that ViP and DP-Cap are both private models pretrained on LAION.
>
> [SSS+2017] Reza Shokri, Marco Stronati, Congzheng Song, Vitaly Shmatikov. Membership Inference Attacks against Machine Learning Models. Proceedings of the IEEE Symposium on Security and Privacy, 2017.

---

> > ### Comment · Reviewer_NLQ1 · 2023-11-22
> > **Happy to raise score**
> >
> > Thanks for the reviewers' detailed responses. Most of my concerns are solved, so i'm happy to raise my score.

---

> > > ### Author Response · Authors · 2023-11-22
> > > **Thanks**
> > >
> > > Thank you. We would be happy to answer any additional question that you might have.

---

### Official Review · Reviewer_fvPE · 2023-11-05

**Soundness:** 2 fair
**Presentation:** 3 good
**Contribution:** 2 fair
**Rating:** 3
**Confidence:** 4

**Summary:**

This paper proposes a DP mechanism for pertaining the Vision-Language foundation model under the image captioning task. The proposed method achieved some markable number, e.g., epsilon=8, the proposed image captioner attains 52.8% accuracy on CIFAR-10. This work shows a pathway for foundation models to be equipped with the DP guarantee.

**Strengths:**

1. The paper is well written and well organized with sufficient background introduction, and the related work analysis.

2. The work shows promising results compared to some representative DP foundation model pretraining, e.g., ViP.

3. There are extensive experiments compared to DP and Non-DP baselines.

**Weaknesses:**

1. The novelty of the paper seems not fully justified. For example, in DP guarantee, what is the difference between DP-Cap and ViP? Under what architecture or updating mechanism design, DP-Cap can claim the novelty or difference?

2. The paper lacks systematic experiment design. Or the motivation of the experiments is not clear.

For example, we observe the foundation model achieve better performance on some vision tasks compared to the original transformer based architecture, which has been observed on ImageNet-1K classification problem. Exactly on this dataset, it should firstly show the traditional classification task compared to other DP methods, and then go for few-shot setting.

Currently, only reporting the ImageNet-1K on few-shot setting with very low numbers (e.g., in Table 3), cannot lead to any positive conclusion.

3. Some experiments comparison cannot draw any conclusion, e.g., Table 2 compared between DP-Cap and the non-DP based methods.

4. One consideration is to replicate the experimental settings from ViP, e.g., DP fine-tuning evaluation on ImageNet-1K, experiments on Caltech-101 and CIFAR-100, which can provide direct comparison to ViP and other DP methods, as appeared in ViP paper. Otherwise, the paper will fail in setting consistent comparison to the literature.

**Questions:**

The main concern comes from the method novelty and the experimental settings. For details, please refer to weakness session.

---

> ### Author Response · Authors · 2023-11-18
> **Rebuttal by Authors (Part 1)**
>
> We thank the reviewer for providing valuable suggestions and comments.
>
> >**Q1**: *"The novelty of the paper seems not fully justified. For example, in DP guarantee, what is the difference between DP-Cap and ViP?"*
>
> **A1**: The novelty of our work lies in building the first differentially private vision-language foundation model, and demonstrating that such models can learn way better private vision encoders compared to previous state-of-the-art DP methods. Regarding the DP guarantee, $\varepsilon$-DP bounds the amount of information extracted from each training sample by $\varepsilon$. Notably, for DP-Cap, each sample is made of {image + caption}, while ViP utilizes {image} only. Consequently, DP-Cap inherently offers an equivalent or better privacy guarantee for each image. One way to see it is to note that DP provides protection against membership inference attacks [SSS+2017]. Suppose $\varepsilon$-DP upper bounds the success rate of a membership inference attack (when given the image-text pair) against DP-Cap as $\leq p$. Then the MIA success rate when given only the image can be at most $p$ since the attacker has strictly less information. This is exactly the upper bound for the success rate of a membership inference attack against ViP. In other words, any attacker that can attack the {image+caption} models (such as DP-Cap) can also attack the image only models (such as ViP).
>
> On the other hand, since the {image+caption} models utilize the caption, the privacy leakage from the text part of the image-caption pair is non-zero for $\varepsilon>0$. It is worth noting that in our set up since we use DP, we protect the captions with the same $\varepsilon$-DP guarantee. Thus, the privacy protection for DP-Cap is neither strictly stronger nor strictly weaker than that for ViP, so the two privacy notions are not directly comparable.
>
> We thank the reviewer for this remark, and are adding this precision in the revised manuscript (Section A.1).
>
> >**Q2**: *"Under what architecture or updating mechanism design, DP-Cap can claim the novelty or difference?"*
>
> **A2**: The vast majority of existing work on DP training operates on a single modality, be it image classification, image-only SSL, text classification, language modeling, etc. Our work is the first to push beyond this barrier and show that DP can also be successfully applied to the currently mainstream paradigm of multimodal learning. We intentionally stuck to well-established and proven techniques for this purpose, using the standard DP-SGD algorithm and Renyi privacy accounting, and the Cap algorithm for image captioning pre-training. By doing so, we hope to establish a strong baseline for DP pre-training on multimodal data, and lay the groundwork for future research by releasing our code/model so that others can build upon our successful training recipe.
>
> Nevertheless, our work does reveal several novel and interesting findings regarding the suitability of image captioning for large-scale DP training, including its ability to scale to massive batch sizes as large as 1.3M, and its synergy with the TAN simulation framework for efficient exploration of training algorithms and hyperparameters. These findings, although not novel in terms of techniques, are crucial for future research since they collectively form a comprehensive strategy for scaling up DP training.
>
>
> [SSS+2017] Reza Shokri, Marco Stronati, Congzheng Song, Vitaly Shmatikov. Membership Inference Attacks against Machine Learning Models. Proceedings of the IEEE Symposium on Security and Privacy, 2017.

---

> ### Author Response · Authors · 2023-11-18
> **Rebuttal by Authors (Part 2)**
>
> >**Q3**: *"Currently, only reporting the ImageNet-1K on few-shot setting with very low numbers (e.g., in Table 3), cannot lead to any positive conclusion. Some experiments comparison cannot draw any conclusion, e.g., Table 2 compared between DP-Cap and the non-DP based methods. One consideration is to replicate the experimental settings from ViP, e.g., DP fine-tuning evaluation on ImageNet-1K, experiments on Caltech-101 and CIFAR-100, which can provide direct comparison to ViP and other DP methods, as appeared in ViP paper. Otherwise, the paper will fail in setting consistent comparison to the literature""*
>
> **A3**: We acknowledge the reviewers' insightful suggestion to bolster the demonstration of DP-Cap's superiority through additional evaluations. In response, we have incorporated fine-tuning experiments into the revised version, Table 9 in Section B.1. As shown in the updated results, DP-Cap improves over ViP on few-shot fine-tuning tasks.
>
> Addressing the concern about the perceived low numbers in few-shot learning, it's crucial to underscore that these figures hold significant importance for benchmarking quality of learned image representations in the self-supervised learning literature, e.g., [ACM+2022]. While the results in Figure 1(a) and Table 4 may appear modest compared with non-private baseline methods, they represent a noteworthy stride towards closing the performance gap with the non private representations as well as improving existing state-of-the-art DP vision foundation models (ViPs). Note that, as detailed in the general comment, we have fixed a bug in the loss scaling which has considerably improved the performance of the vision encoder across all benchmarks. For instance, we now achieve a performance of 63.4% on ImageNet full linear probing—a nearly 8% improvement over ViP.
>
>
> >**Q4**: *“Some experiments comparison cannot draw any conclusion, e.g., Table 2 compared between DP-Cap and the non-DP based methods.”*
>
> **A4**: It is a common practice to report the non-private model performance in the differential privacy literature, which provides information about the performance gap between private models and non-private models. Thank you for your question, we have added a remark on the non-private models to improve presentation clarity in Section B1.
>
> We hope that we have adequately addressed the queries posed by the reviewer. We are readily available for any further discussions or clarifications as needed.
>
> [ACM+2022] Masked siamese networks for label-efficient learning. Mahmoud Assran, Mathilde Caron, Ishan Misra, Piotr Bojanowski, Florian Bordes, Pascal Vincent, Armand Joulin, Michael Rabbat, Nicolas Ballas. ECCV 2022.

---

> > ### Author Response · Authors · 2023-11-22
> > **End of author-reviewer discussion period**
> >
> > Dear reviewer, please find bellow the markdown version of the fine-tuning evaluation you have asked for.  Note that the table was placed in appendix B.1.
> >
> > | Model      |           | AirCraft  |           |           | Caltech-101 |           |           | Cifar-100 |           |
> > |------------|-----------|-----------|-----------|-----------|-------------|-----------|-----------|-----------|-----------|
> > |            | 10-shot   | 20-shot   | 30-shot   | 5-shot    | 10-shot     | 30-shot   | 5-shot    | 10-shot   | 30-shot   |
> > | AlexNet    | 23.3%     | 34.4%     | 41.4%     | 64.7%     | 73.6%       | 81.4%     | 29.7%     | 36.3%     | 49.3%     |
> > | SimCLR     | 38.8%     | 56.9%     | 64.9%     | 81.7%     | 89.1%       | 94.5%     | 49.9%     | 60.2%     | 71.8%     |
> > | TAN        | 22.8%     | 37.9%     | 46.0%     | 49.3%     | 66.4%       | 77.9%     | 21.3%     | 27.8%     | 42.4%     |
> > | ViP        | 31.6%     | 53.1%     | 64.3%     | 68.1%     | 79.0%       | 88.9%     | 30.7%     | 41.0%     | 57.5%     |
> > |------------|-----------|-----------|-----------|-----------|-------------|-----------|-----------|-----------|-----------|
> > | **DP-Cap** | **37.5%** | **57.9%** | **66.7%** | **70.3%** | **81.3%**   | **90.0%** | **36.3%** | **46.3%** | **62.1%** |
> >
> > The choice for the models is the same as for table1: AlexNet and SinCLR are non private, while TAN/ViP are also trained under $\varepsilon=8$.
> >
> > We would be very happy to answer any additional question that the reviewer might have.

---

### Author Response · Authors · 2023-11-18
**Message to all reviewers**

We thank the reviewers for their insightful comments and questions. We have responded to each reviewer’s question, and updated a revision of the paper to take these comments into account. We've for instance incorporated a fine-tuning evaluation of our models in appendix B.1.

After paper submission, we discovered a bug that affected hyperparameter optimality: we were downscaling the loss before performing the backpropagation. It in no way affected the correctness of the DP guarantees, as loss scaling happened before gradient clipping. However, correcting this changed several things:

- It brought a notable performance boost across all benchmarks. The updated results are visually highlighted in blue for clarity, and also modified in the figures. For instance, our model now achieves a linear probing accuracy of 63.4% on ImageNet under $\varepsilon=8$, marking a significant improvement by adding 5 points compared to our previous result. The improvement is consistently observed across all the evaluations of our vision backbones, and it also enhanced the zero-shot performance. We hope that it will further convince the reviewers on the strength of our models. We now also observe that the ViT-base model performs better than the small, suggesting that model scaling can be effective for DP VLM pre-training.

- The scaling properties concerning batch size, as previously depicted in Figure 3.a, differ from what we initially observed: Fixing the bug also considerably improved the scaling laws of both ViP and DP-Cap, which means that both models do not seem to suffer from the performance drop observed in [SSS2023] as a result of poor optimization. To better understand the relative advantage of large batch training for DP-Cap compared to ViP, we have trained a new DP-Cap model with the exact same privacy-parameters as the original ViP (notably batch size 97k instead of 1.3M), and a ViP model with the same parameters as our DP-Cap. We observe that the improvement between 90k to 1.3M is almost negligible for the reconstruction-based method (ViP), while it is crucially beneficial to DP-Cap. We have added a new plot in figure 3.b that shows how this scaling benefits DP-Cap much more than ViP.

On a minor note, the models are now re-trained on a modified version of our DEDUP-LAION-233M dataset, where faces have been intentionally blurred. This dataset remains identical to its non-blurred counterpart, with the sole distinction of facial blurring, which was done in a similar manner as [YYF+2022]. We updated some images and generated captions in the paper to align with either the top 10% of captions based on CIDEr score (Figure 1 and 7) or are randomly selected (Figure 6).

Overall, stemming from the realization that image descriptions allow models to focus on crucial objects and their relationships, we found it particularly advantageous for DP where models must adeptly extract essential information within an $\varepsilon$ constraint from each sample. Our demonstrations reveal DP-Cap significantly outperforming prior DP vision encoders, and introduces multi-modal capabilities—a first for DP. Furthermore, we've unearthed critical insights, notably the "dilemma of large batch training in DP-SGD." This arises from balancing a robust theoretical guarantee ($\varepsilon$) with minimal noise per gradient update, achievable only through substantial batch sizes. Successfully scaling to a batch size of 1.3 million proved pivotal for our model's success. However, our investigations revealed that other models can not leverage the good SNR achieved with DP-Cap.

[YYF+2022] A Study of Face Obfuscation in ImageNet. Kaiyu Yang, Jacqueline Yau,  Li Fei-Fei, Jia Deng, Olga Russakovsky. ICML 2022.

[SSS2023] TAN Without a Burn: Scaling Laws of DP-SGD. Tom Sander, Pierre Stock, Alexandre Sablayrolles. ICML 2023.

---

### Author Response · Authors · 2023-11-20
**Rebuttal Deadline Approaching**

Dear Reviewers, Area Chairs and Program Chairs,

We extend our sincere gratitude for the reviewers’ valuable comments and questions. We have strived to address them with utmost precision, refining our article accordingly.

As a gentle reminder, the author-reviewers discussion period concludes on Wednesday (November 22nd). We haven’t gotten any response so far; we want to reiterate that we would love to engage in more discussion. If further clarification is needed, we remain readily available and eager to address any queries related to our work.

Best regards,

The authors

---

### Meta-Review · Area_Chair_D83A · 2023-12-05

**Metareview:**

The paper presents a novel approach for learning differentially private vision-foundation models via image captioning. The key innovation of the method is the use of an image captioning objective that enables the direct application of DP-SGD, since the loss is an additive function over the samples. Compared to the existing approach of using a masked autoencoder objective, the proposed image caption objective not only enables vision-language tasks, but allows for the use of much larger batch sizes during model training, without degrading the model performance, which greatly reduces the effective noise per training iteration, leading to better privacy-utility tradeoffs. In experimental evaluations on downstream vision and vision-language tasks, the method achieves SoTA performance relative to DP baselines and, on some tasks, even comparable performance to non-private models. Specifically, the strength of this paper includes several aspects. 1)On the linear probing classification, the proposed method performs comparably or better than non-private AlexNet and SimCLR, and meaningfully outperforms the SoTA DP baseline ViP. 2) On few-shot learning, the proposed method more than doubles the accuracy of the SoTA baseline ViP. 3) On ARO, the proposed method performs comparably or better than non-private CLIP.

However, there are several points to be further improved. For example, this paper lacks any meaningful theoretical or algorithmic contributions. The only insight of the paper is that image captioning is a suitable loss for DP training. The technical novelty of the proposed method is somehow limited. To be specific, the proposed method consists of two training stages. The first stage trains the model on texture images similarly as proposed in Yu er al. (2023). Next, the second stage further trains the model on image captioning task under the DP framework, which is quite straightforward. Except for DP, is there any other frameworks/methods for overcoming the copyright/privacy issue? It is clear from the manuscript that DP-based models performs much inferior to non-DP-based models, yet it is not clear is there any alternative to DP. Therefore, this paper cannot be accepted at ICLR this time, but the enhanced version is highly encouraged to submit other top-tier venues.

**Justification For Why Not Higher Score:**

However, there are several points to be further improved. For example, this paper lacks any meaningful theoretical or algorithmic contributions. The only insight of the paper is that image captioning is a suitable loss for DP training. The technical novelty of the proposed method is somehow limited. To be specific, the proposed method consists of two training stages. The first stage trains the model on texture images similarly as proposed in Yu er al. (2023). Next, the second stage further trains the model on image captioning task under the DP framework, which is quite straightforward. Except for DP, is there any other frameworks/methods for overcoming the copyright/privacy issue? It is clear from the manuscript that DP-based models performs much inferior to non-DP-based models, yet it is not clear is there any alternative to DP. Therefore, this paper cannot be accepted at ICLR this time, but the enhanced version is highly encouraged to submit other top-tier venues.

**Justification For Why Not Lower Score:**

However, there are several points to be further improved. For example, this paper lacks any meaningful theoretical or algorithmic contributions. The only insight of the paper is that image captioning is a suitable loss for DP training. The technical novelty of the proposed method is somehow limited. To be specific, the proposed method consists of two training stages. The first stage trains the model on texture images similarly as proposed in Yu er al. (2023). Next, the second stage further trains the model on image captioning task under the DP framework, which is quite straightforward. Except for DP, is there any other frameworks/methods for overcoming the copyright/privacy issue? It is clear from the manuscript that DP-based models performs much inferior to non-DP-based models, yet it is not clear is there any alternative to DP. Therefore, this paper cannot be accepted at ICLR this time, but the enhanced version is highly encouraged to submit other top-tier venues.

---

### Decision · Program_Chairs · 2024-01-16

Reject